# FakeShield: Explainable Image Forgery Detection and Localization via Multi-modal Large Language Models

**Zhipei Xu**[1,2]**, Xuanyu Zhang**[1]**, Runyi Li**[1]**, Zecheng Tang**[1]**, Qing Huang**[4]**, Jian Zhang**[1,2,3†]

[1]School of Electronic and Computer Engineering, Peking University
[2]Peking University Shenzhen Graduate School-Rabbitpre AIGC Joint Research Laboratory
[3]Guangdong Provincial Key Laboratory of Ultra High Definition Immersive Media Technology,
Shenzhen Graduate School, Peking University
[4]School of Future Technology, South China University of Technology

## Abstract

The rapid development of generative AI is a double-edged sword, which not only facilitates content creation but also makes image manipulation easier and more difficult to detect. Although current image forgery detection and localization (IFDL) methods are generally effective, they tend to face two challenges: **1)** black-box nature with unknown detection principle, **2)** limited generalization across diverse tampering methods (e.g., Photoshop, DeepFake, AIGC-Editing). To address these issues, we propose the explainable IFDL task and design FakeShield, a multimodal framework capable of evaluating image authenticity, generating tampered region masks, and providing a judgment basis based on pixel-level and image-level tampering clues. Additionally, we leverage GPT-4o to enhance existing IFDL datasets, creating the Multi-Modal Tamper Description dataSet (MMTD-Set) for training FakeShield's tampering analysis capabilities. Meanwhile, we incorporate a Domain Tag-guided Explainable Forgery Detection Module (DTE-FDM) and a Multi-modal Forgery Localization Module (MFLM) to address various types of tamper detection interpretation and achieve forgery localization guided by detailed textual descriptions. Extensive experiments demonstrate that FakeShield effectively detects and localizes various tampering techniques, offering an explainable and superior solution compared to previous IFDL methods. The code is available at `https://github.com/zhipeixu/FakeShield`.

## 1 Introduction

With the rapid development of AIGC, powerful image editing models have provided a breeding ground for convenient image tampering, blurring the boundaries between true and forgery. People can use AIGC image editing methods (Rombach et al., 2022; Zhang et al., 2023; Suvorov et al., 2022; Mou et al., 2023) to edit images without leaving a trace. Although it has facilitated the work of photographers and illustrators, AIGC editing methods have also led to an increase in malicious tampering and illegal theft. The authenticity of images in social media is difficult to guarantee, which will lead to problems such as rumor storms, economic losses, and legal concerns. Therefore, it is important and urgent to identify the authenticity of images. In this context, the image forgery detection and localization (IFDL) task aims to identify whether an image has been tampered with and locate the specific manipulation areas. It can be widely applied in the real world, such as filtering false content on social media, preventing the spread of fake news, and court evidence collection.

State-of-the-art IFDL methods have utilized well-designed network structures, elaborate network constraints, and efficient pre-training strategies to achieve remarkable performance (Yu et al., 2024b; Ma et al., 2023; Dong et al., 2022). However, previous IFDL methods face two key problems, lim-

† Corresponding author: Jian Zhang. This work was supported in part by Guangdong Provincial Key Laboratory of Ultra High Definition Immersive Media Technology (No. 2024B1212010006) and Shenzhen General Research Project (No. JCYJ20241202125904007).

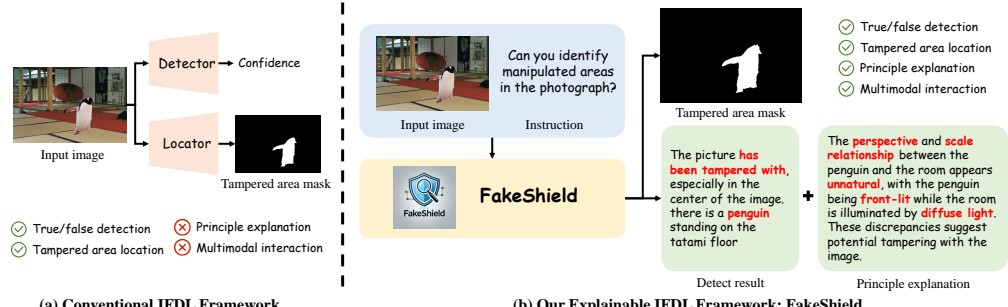

Figure 1: Illustration of the conventional IFDL and explainable IFDL framework. Conventional methods offer only detection results and tampered masks. We extend this into a multi-modal framework, enabling detailed explanations and conversational interactions for a deeper analysis.

iting their practicality and generalizability. **First**, as shown in Figure 1(a), most existing IFDL methods are black-box models, only providing the authenticity probability of the image, while the principle of detection is unknown to users. Since the existing IFDL methods cannot guarantee satisfactory accuracy, manual subsequent judgment is still required. Given that the information provided by the IFDL methods is insufficient, it is difficult to support the human assessment and users still need to re-analyze the suspect image by themselves. **Second**, in real-world scenarios, tampering types are highly diverse, including Photoshop (copy-and-move, splicing, and removal), AIGC-Editing, DeepFake, and so on. Existing IFDL methods (Yu et al., 2024b; Ma et al., 2023) are typically limited to handling only one of these techniques, lacking the ability to achieve comprehensive generalization. This forces users to identify different tampering types in advance and apply specific detection methods accordingly, significantly reducing these models' practical utility.

Benefiting from the rapid advancements in Transformer architectures, Large Language Models (LLMs) have attracted significant attention. Furthermore, (Liu et al., 2024) introduced a Multimodal Large Language Model (M-LLM) that aligns visual and textual features, thereby endowing LLMs with enhanced visual comprehension abilities. Given that LLMs are pre-trained on an extensive and diverse corpus of world knowledge, they hold significant potential for a wide range of applications, such as machine translation (Devlin, 2018), code completion, and visual understanding (Liu et al., 2024). Consequently, we explored the feasibility of employing M-LLMs for explainable Image Forgery Detection and Localization (e-IFDL). This approach allows for a more comprehensive explanation of the rationale behind tampering detection and provides a more precise identification of both the authenticity of images and the suspected manipulation regions.

To address the two issues of the existing IFDL methods, we propose the explainable-IFDL (e-IFDL) task and a multi-modal explainable tamper detection framework called FakeShield. As illustrated in Figure1(b), the e-IFDL task requires the model to evaluate the authenticity of any given image, generate a mask for the suspected tampered regions, and provide a rationale based on some pixel-level artifact details (e.g., object edges, resolution consistency) and image-level semantic-related errors (e.g., physical laws, perspective relationships). Leveraging the capabilities of GPT-4o (OpenAI, 2023), we can generate a comprehensive triplet consisting of a tampered image, a modified area mask, and a detailed description of the edited region through a meticulously crafted prompt. Then, we develop the **M**ulti-Modal **T**amper **D**escription data**Set** (MMTD-Set) by building upon existing IFDL datasets. Utilizing the MMTD-Set, we fine-tune M-LLM (Liu et al., 2024) and visual segmentation models (Kirillov et al., 2023; Lai et al., 2024), equipping them with the capability to provide complete analysis for judgment, detecting tampering, and generate accurate tamper area masks. This process ultimately forms a comprehensive forensic pipeline for analysis, detection, and localization. Our contributions are summarized as follows:

❑ (1) We present the first attempt to propose a multi-modal large image forgery detection and localization model, dubbed **FakeShield**. It can not only decouple the detection and localization process but also provide a reasonable judgment basis, which alleviates the black-box property and unexplainable issue of existing IFDL methods.

❑ (2) We use GPT-4o to enrich the existing IFDL dataset with textual information, constructing the **MMTD-Set**. By guiding it to focus on distinct features for various types of tampered data, GPT-4o can analyze the characteristics of tampered images and construct "image-mask-description" triplets.

❏ (3) We develop a **D**omain **T**ag-guided **E**xplainable **F**orgery **D**etection **M**odule (**DTE-FDM**) to spot different types of fake images in a united model and effectively alleviate the data domain conflict. Meanwhile, an **M**ulti-modal **F**orgery **L**ocalization **M**odule (**MFLM**) is adopted to align visual-language features, thus pinpointing tampered areas.

❏ (4) Extensive experiments demonstrate that our method can accurately analyze tampering clues, and surpass most previous IFDL methods in the detection and localization of many tampering types like copy-move, splicing, removal, DeepFake, and AIGC-based editing.

## 2 RELATED WORKS

### 2.1 IMAGE FORGERY DETECTION AND LOCALIZATION

Prevailing IFDL methods mainly target at the localization of specific manipulation types (Salloum et al., 2018; Islam et al., 2020; Li & Zhou, 2018; Zhu et al., 2018; Li & Huang, 2019). In contrast, universal tamper localization methods (Li et al., 2018; Kwon et al., 2021; Chen et al., 2021; Ying et al., 2023; 2021; Hu et al., 2023; Ying et al., 2022; Li et al., 2024; Yu et al., 2024a; Zhang et al., 2024b) aim to detect artifacts and irregularities across a broader spectrum of tampered images. For instance, MVSS-Net (Dong et al., 2022) utilized multi-scale supervision and multi-view feature learning to simultaneously capture image noise and boundary artifacts. OSN (Wu et al., 2022) employed a robust training strategy to overcome the difficulties associated with lossy image processing. HiFi-Net (Guo et al., 2023) adopted a combination of multi-branch feature extraction and localization modules to effectively address alterations in images synthesized and edited by CNNs. IML-ViT (Ma et al., 2023) integrated Swin-ViT into the IFDL task, employing an FPN architecture and edge loss constraints to enhance its performance. DiffForensics (Yu et al., 2024b) adopted a training approach akin to diffusion models, strengthening the model's capacity to capture fine image details. Additionally, some researchers (Zhang et al., 2024a;c; Asnani et al., 2023) have pursued proactive tamper detection and localization by embedding copyright and location watermarks into images/audio/videos preemptively. However, despite their acceptable performances, these IFDL methods cannot explain the underlying principles and rationale behind their detection and localization judgments, offering no interaction. Moreover, they suffer from limited generalization and accuracy, exhibiting significant performance disparities across different testing data domains.

### 2.2 LARGE LANGUAGE MODEL

Large language models (Dubey et al., 2024; OpenAI, 2023) have garnered global attention in recent years for their exceptional instruction-following and text-generation abilities. Based on the Transformer architecture, LLMs are pre-trained on massive datasets, allowing them to accumulate broad world knowledge that enhances their ability to generalize across a wide range of downstream tasks. Subsequently, some researchers (Li et al., 2022) expanded LLMs' powerful understanding and world knowledge to the visual domain by incorporating image encoders and projection layers, which enable images encoded into tokens that align with the text. Some recent works (Chen et al., 2023a; Wang et al., 2023; Chen et al., 2023b) equipped M-LLMs with enhanced visual understanding capabilities by expanding the visual instruction datasets and increasing the model size during fine-tuning. Currently, M-LLMs demonstrate impressive performance across various downstream tasks. LISA (Lai et al., 2024) integrated SAM (Kirillov et al., 2023) with M-LLM to implement reasoning segmentation, enabling the generation of masks from text descriptions. GLaMM (Rasheed et al., 2024) further enhanced this by using a more advanced region image encoder to improve text-to-mask grounding. Additionally, some studies (Yang & Zhou, 2024; Zhang et al., 2024d) have explored the application of M-LLMs in DeepFake detection. For instance, (Zhang et al., 2024d) introduced the DD-VQA dataset, combining a manual inference process for rating real and fake faces that can be distinguished using common sense. Targeted at Deepfake detection, (Huang et al., 2024) used GPT-4o to create image-analysis pairs, and introduced a multi-answer intelligent decision system into MLLM, achieving good effect. However, it cannot be generalized to other types of tampering such as Photoshop and AIGC-Editing, and cannot accurately locate the tampered areas. Besides, using M-LLMs to realize universal tamper localization and detection remains unexplored.

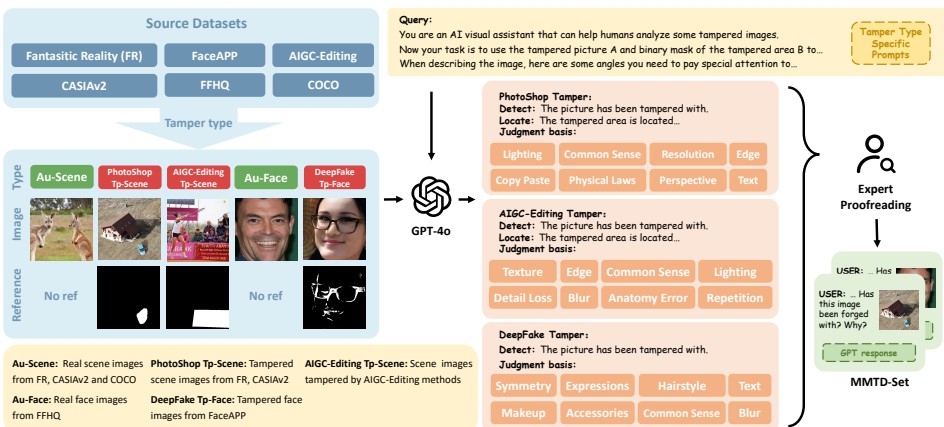

Figure 2: Illustration of the construction process of our MMTD-Set. We sample the tampered image-mask pairs from PS, DeepFake, and AIGC benchmarks, and then use domain tags to guide GPT-4o in constructing the judgment basis and focusing on both pixel-level details and image-level content.

## 3 METHODOLOGY

### 3.1 CONSTRUCTION OF THE PROPOSED MMTD-SET

**Motivation:** Most existing IFDL datasets consist of a single visual modality, lacking training visual-language samples adapted to M-LLMs. The challenge of constructing our MMTD-Set lies in accurately translating the visual tampering information from the existing IFDL image datasets into precise textual descriptions. To address this challenge, our core contributions focus on two aspects: **(1)** We leverage GPT-4o to generate text description and provide both the tampered image and its corresponding mask to GPT-4o, enabling it to accurately identify the tampered location. **(2)** For each tamper type, we design specific prompts to their unique characteristics, guiding GPT-4o to focus on different tampering artifacts and providing more detailed visual cues.

**Data collection:** Based on (Ma et al., 2023; Nirkin et al., 2021), we categorize common tampering into three types: PhotoShop (copy-move, splicing, removal), DeepFake (FaceAPP (FaceApp Limited, 2017)), and AIGC-Editing (SD-inpainting (Lugmayr et al., 2022)). As shown in Figure 2, we gathered three types of tampered images along with their corresponding authentic images from public datasets (Dong et al., 2013; Dang et al., 2020) and self-constructed data.

**GPT assisted description generation:** Given that manual analysis of tampered images is time-consuming, inspired by (Liu et al., 2024; Chen et al., 2023a; Huang et al., 2024), we used GPT-4o to automate the analysis of tampered images. As depicted in Figure 2, the output analysis is required to follow the format of detected results, localization descriptions, and judgment basis.

**For tampered images**, we input the edited image, its corresponding forgery mask, and our carefully constructed tamper type specific prompts into the powerful GPT-4o to more accurately describe the tampered regions. **For authentic images**, GPT-4o is provided with only the real image and a set of prompts, guiding it to confirm its authenticity. *The full-text prompts are detailed in the Appendix A.6.* To more clearly and specifically describe and analyze the tampering of images, GPT-4o describes the image from two key aspects: the location and content of the tampered areas, and any visible artifacts or semantic errors caused by the tampering: **(1)** For the tampering location, GPT-4o is required to describe it in both absolute positions (e.g., top, bottom, upper left corner, lower right corner) and relative positions (e.g., above the crowd, on the table, under the tree). When analyzing the tampered content, it is tasked with providing detailed information about the types, quantities, actions, and attributes of the objects within the tampered region. **(2)** For the visible artifacts and semantic errors, since different tampering methods produce distinct types of artifacts, we craft specific prompts to guide the analysis. It can broadly be categorized into pixel-level artifact details and image-level semantic-related errors. For PhotoShop (PS) tampering, operations like copy-move and splicing often introduce pixel-level issues such as edge artifacts, abnormal resolution, and inconsistencies in lighting. Additionally, semantic-level errors, including violations of physical laws or common sense, are frequently observed. In AIGC-Editing (AIGC), for instance, it often fails to generate text

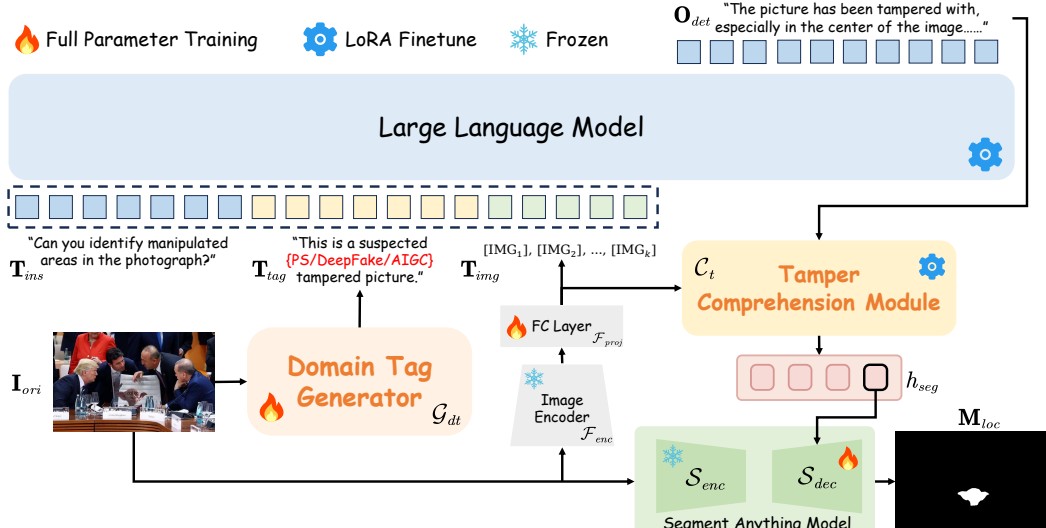

Figure 3: The pipeline of FakeShield. Given an image $\mathbf{I}_{ori}$ for detection, it is first processed by the Domain Tag Generator $\mathcal{G}_{dt}$ to obtain a data domain tag $\mathbf{T}_{tag}$. The tag $\mathbf{T}_{tag}$, along with the text instruction $\mathbf{T}_{ins}$ and image tokens $\mathbf{T}_{img}$, are simultaneously input into the fine-tuned LLM, generating tamper detection result and explanation $\mathbf{O}_{det}$. Subsequently, $\mathbf{O}_{det}$ and $\mathbf{T}_{img}$ are input into the Tamper Comprehension Module $\mathcal{C}_t$, and the last-layer embedding for the <SEG> token $\mathbf{h}_{\texttt{<SEG>}}$ serves as a prompt for SAM, guiding it to generate the tamper area mask $\mathbf{M}_{loc}$.

accurately, resulting in disordered symbols or characters appearing in the tampered area. For the DeepFake (DF), tampering with facial features frequently results in localized blurring.

## 3.2 OVERALL FRAMEWORK OF FAKESHIELD

Our goals involve two issues: **1):** Utilizing the textual understanding ability and world knowledge of the M-LLM to analyze and judge the authenticity of images; **2):** Adopting the analysis and interpretation of tampered images to assist the segmentation model in pinpointing the tampered areas. To solve these two tasks, an intuitive approach is to fine-tune a large multimodal model to simultaneously output analysis and tampered masks. However, we find that joint training of multiple tasks will increase the difficulty of network optimization and interfere with each other. Considering that detection and interpretation focus more on language understanding and organization, while localization requires more accumulation of visual prior information, the proposed FakeShield contains two key decoupled parts, namely DTE-FDM and MFLM, as illustrated in Fig. 3. Specifically, an original suspected image $\mathbf{I}_{ori}$ and an instruction text $\mathbf{T}_{ins}$ (e.g. "Can you identify manipulated areas in the photograph?") are fed to the proposed DTE-FDM to predict the detection result and judgment basis $\mathbf{O}_{det}$. In this process, we use a learnable generator to produce a domain tag $\mathbf{T}_{tag}$, thus avoiding the tampered data domain conflict. Furthermore, we input the interpretation $\mathbf{O}_{det}$ and the image $\mathbf{I}_{ori}$ to the MFLM to accurately extract the tampered mask $\mathbf{M}_{loc}$. To promote cross-modal interaction for tamper localization, we introduce a tamper comprehension module to align the visual and textual features and enhance the ability of the vision foundation model to understand long descriptions.

## 3.3 DOMAIN TAG-GUIDED EXPLAINABLE FORGERY DETECTION MODULE

**Motivation:** In real-life scenarios, images can be tampered with and attacked through various methods, including copy-move, splicing, removal, DeepFake, and AIGC-based methods. However, these tampered images have different distribution characteristics, and domain differences, making it difficult to apply a single IFDL method to all forgery data. For example, DeepFake focuses on face modification, often causing partial blurring and unnatural features in the lips, teeth, and eyes. In contrast, tools like PhotoShop (splicing, copy-move, removal) tend to leave noticeable artifacts at the edges of the tampered areas. In the case of AIGC-Editing, blurring within the tampered region often alters or obscures texture details. To mitigate these significant domain discrepancies, inspired

by (Sanh et al., 2022), we introduce the **Domain Tag Generator(DTG)**, which utilizes a specialized domain tag to prompt the model to distinguish between various data domains.

First, the original image $\mathbf{I}_{ori}$ is input into a classifier $\mathcal{G}_{dt}$ to obtain the domain tag $\mathbf{T}_{tag}$. Specifically, we classify all common tampering types into three categories: Photoshop-based editing, DeepFake, and AIGC-based tampering, and use the template *"This is a suspected {data domain}-tampered picture."* as the identifier. Simultaneously, consistent with (Liu et al., 2024), $\mathbf{I}_{ori}$ is passed through the image encoder $\mathcal{F}_{enc}$ and linear projection layer $\mathcal{F}_{proj}$ to generate the image tokens [IMG] $\mathbf{T}_{img}$. Next, $\mathbf{T}_{tag}$ and $\mathbf{T}_{img}$ are concatenated with the instruction $\mathbf{T}_{ins}$ and then fed into the LLM. To be noted, $\mathbf{T}_{ins}$ is a prompt that instructs the model to detect tampering and describe the location of the manipulation, for example: *"Can you identify manipulated areas in the photograph?"*. After several autoregressive predictions, the output $\mathbf{O}_{det}$ comprises three components: detection results, a description of the location of tampered area, and the interpretive basis for the detection.

$$\mathbf{T}_{tag} = \mathcal{G}_{dt}(\mathbf{I}_{ori}), \quad \mathbf{T}_{img} = \mathcal{F}_{proj}(\mathcal{F}_{enc}(\mathbf{I}_{ori})) \tag{1}$$

$$\mathbf{O}_{det} = \mathrm{LLM}(\mathbf{T}_{ins}, \mathbf{T}_{tag} \mid \mathbf{T}_{img}). \tag{2}$$

Given the large size of LLMs and limited computational resources, full parameter training is impractical. Thus, we freeze the LLM and leverage LoRA fine-tuning technology (Hu et al., 2022) to preserve semantic integrity while enabling efficient image forgery detection.

### 3.4 MULTI-MODAL FORGERY LOCALIZATION MODULE

**Motivation:** Although $\mathbf{O}_{det}$ provides a textual description of the tampered area, it lacks precision and intuitive clarity. To address this issue, we aim to transform $\mathbf{O}_{det}$ into an accurate binary mask, providing a clearer and more accurate representation of the tampered region. Existing prompt-guided segmentation algorithms (Kirillov et al., 2023; Lai et al., 2024) struggle to capture the semantics of long texts and hard to accurately delineate modified regions based on detailed descriptions. Inspired by (Lai et al., 2024), we propose a **Tamper Comprehension Module (TCM)**, which is an LLM serving as an encoder aligns long-text features with visual modalities, enhancing SAM's precision in locating the forgery areas. To generate the prompt fed into SAM, following (Lai et al., 2024), we introduce a specialized token <SEG>.

As shown in Fig.3, the tokenized image $\mathbf{T}_{img}$ and the tampered description $\mathbf{O}_{det}$ are fed into the TCM $\mathcal{C}_t$. Then, we extract the last-layer embedding of TCM and transform it into $\mathbf{h}_{<SEG>}$ via an MLP projection layer. Simultaneously, the original image $\mathbf{I}_{ori}$ is processed through the SAM encoder $\mathcal{S}_{enc}$ and decoder $\mathcal{S}_{dec}$, where $\mathbf{h}_{<SEG>}$ serve as a prompt for $\mathcal{S}_{dec}$ guiding the mask generation $\mathbf{M}_{loc}$.

$$\mathbf{E}_{mid} = \mathcal{S}_{enc}(\mathbf{I}_{ori}), \quad \mathbf{h}_{<SEG>} = \mathrm{Extract}(\mathcal{C}_t(\mathbf{T}_{img}, \mathbf{O}_{det}))$$
$$\mathbf{M}_{loc} = \mathcal{S}_{dec}(\mathbf{E}_{mid} \mid \mathbf{h}_{<SEG>}), \tag{3}$$

where $\mathbf{E}_{mid}$ represents the intermediate features of SAM, and $\mathrm{Extract}(\cdot)$ denotes the operation of extracting the last-layer embedding corresponding to the <SEG> token. Similar to DTE-FDM, we also apply LoRA fine-tuning to MFLM for greater efficiency. With the integration of TCM, SAM will achieve more precise localization of the forgery areas.

### 3.5 TRAINING OBJECTIVES

The two submodules of our FakeShield are trained end-to-end separately. **For DTE-FDM**, the domain tag generator utilizes cross-entropy loss $\ell_{ce}$ as its training objective, enabling it to distinguish between different data domains. Following the approach of LLaVA, our LLM's training objective is the cross-entropy loss $\ell_{ce}$. The training target of DTE-FDM $\ell_{det}$ can be formulated as:

$$\ell_{det} = \ell_{ce}(\hat{\mathbf{O}}_{det}, \mathbf{O}_{det}) + \lambda \cdot \ell_{ce}(\hat{\mathbf{T}}_{tag}, \mathbf{T}_{tag}), \tag{4}$$

where $\lambda$ denotes the weight balancing different loss components, $\mathbf{O}_{det}$ and $\mathbf{T}_{tag}$ represent the predictions of LLM and DTG, while $\hat{\mathbf{O}}_{det}$ and $\hat{\mathbf{T}}_{tag}$ represent their corresponding ground truth. **For MFLM**, we apply $\ell_{ce}$ to constrain TCM to produce high-quality prompt $\mathbf{y}_{txt}$ with <SEG> token. Meanwhile, we use a linear combination of binary cross-entropy loss $\ell_{bce}$ and dice loss $\ell_{dice}$ to encourage the output of MFLM $\mathbf{M}_{loc}$ to be close to the GT mask $\hat{\mathbf{M}}_{loc}$. Given the ground-truth prompt $\hat{\mathbf{y}}_{txt}$ (e.g., "It is <SEG>") and mask $\hat{\mathbf{M}}_{loc}$, our training losses for MFLM $\ell_{loc}$ can be formulated as:

$$\ell_{loc} = \ell_{ce}(\hat{\mathbf{y}}_{txt}, \mathbf{y}_{txt}) + \alpha \cdot \ell_{bce}(\hat{\mathbf{M}}_{loc}, \mathbf{M}_{loc}) + \beta \cdot \ell_{dice}(\hat{\mathbf{M}}_{loc}, \mathbf{M}_{loc}), \tag{5}$$

Table 1: Detection performance comparison between our FakeShield and other competitive methods. Our method can achieve the best detection accuracy in PhotoShop, DeepFake, and AIGC-Editing tampered datasets. The best score is highlighted in **bold** and the second-best score is underlined.

| Method | CASIA1+ | | IMD2020 | | PhotoShop Columbia | | Coverage | | DSO | | DeepFake | | AIGC-Editing | |
|---|---|---|---|---|---|---|---|---|---|---|---|---|---|---|
| | ACC | F1 | ACC | F1 | ACC | F1 | ACC | F1 | ACC | F1 | ACC | F1 | ACC | F1 |
| SPAN | 0.60 | 0.44 | 0.70 | 0.81 | 0.87 | 0.93 | 0.24 | 0.39 | 0.35 | 0.52 | 0.78 | 0.78 | 0.47 | 0.05 |
| ManTraNet | 0.52 | 0.68 | 0.75 | 0.85 | 0.95 | 0.97 | 0.95 | 0.97 | 0.90 | 0.95 | 0.50 | 0.67 | 0.50 | 0.67 |
| HiFi-Net | 0.46 | 0.44 | 0.62 | 0.75 | 0.68 | 0.81 | 0.34 | 0.51 | 0.96 | 0.98 | 0.56 | 0.61 | 0.49 | 0.42 |
| PSCC-Net | 0.90 | 0.89 | 0.67 | 0.78 | 0.78 | 0.87 | 0.84 | 0.91 | 0.66 | 0.80 | 0.48 | 0.58 | 0.49 | 0.65 |
| CAT-Net | 0.88 | 0.87 | 0.68 | 0.79 | 0.89 | 0.94 | 0.23 | 0.37 | 0.86 | 0.92 | 0.85 | 0.84 | 0.82 | 0.81 |
| MVSS-Net | 0.62 | 0.76 | 0.75 | 0.85 | 0.94 | 0.97 | 0.65 | 0.79 | 0.96 | 0.98 | 0.84 | 0.91 | 0.44 | 0.24 |
| FakeShield | **0.95** | **0.95** | **0.83** | **0.90** | **0.98** | **0.99** | **0.97** | **0.98** | **0.97** | **0.98** | **0.93** | **0.93** | **0.98** | **0.99** |

where $\alpha$ and $\beta$ are weighting factors used to balance the respective losses. $\ell_{ce}$, $\ell_{bce}$, and $\ell_{dice}$ refer to cross-entropy loss, binary cross-entropy loss, and dice loss (Sudre et al., 2017) respectively.

## 4 EXPERIMENT

### 4.1 EXPERIMENTAL SETUP

**Dataset:** We employ the dataset construction method outlined in Section 3.1 to build the training and test sets of the MMTD-Set. For the training set, we utilize PhotoShop tampering (e.g., CASIAv2 (Dong et al., 2013), Fantastic Reality (Kniaz et al., 2019)), DeepFake tampering (e.g., FFHQ, FaceApp (Dang et al., 2020)), and some self-constructed AIGC-Editing tampered data as the source dataset. For the testing set, we select several challenging public benchmark datasets including PhotoShop tampering (CASIA1+ (Dong et al., 2013), Columbia (Ng et al., 2009), IMD2020 (Novozamsky et al., 2020), Coverage (Wen et al., 2016), DSO (De Carvalho et al., 2013), Korus (Korus & Huang, 2016)), DeepFake tampering (e.g., FFHQ, FaceApp (Dang et al., 2020), Seq-DeepFake (Shao et al., 2022)), and some self-generated AIGC-Editing data.

**State-of-the-Art Methods:** To ensure a fair comparison, we select competitive methods that provide either open-source code or pre-trained models. To evaluate the **IFDL performance** of FakeShield, we compare it against SPAN (Hu et al., 2020), MantraNet (Wu et al., 2019), OSN (Wu et al., 2022), HiFi-Net (Guo et al., 2023), PSCC-Net (Liu et al., 2022), CAT-Net (Kwon et al., 2021), and MVSS-Net (Dong et al., 2022), all of which are retrained on the MMTD-Set for consistency with the same training setup. For **DeepFake detection**, CADDM (Dong et al., 2023), HiFi-DeepFake (Guo et al., 2023), RECCE (Cao et al., 2022) and Exposing (Ba et al., 2024) are chosen as comparison methods. Additionally, to assess the **explanation ability** of FakeShield, we compare it with open-source M-LLMs such as LLaVA-v1.6-34B (Liu et al., 2024), InternVL2-26B (Chen et al., 2024), and Qwen2-VL-7B (Wang et al., 2024), as well as the closed-source model GPT-4o (OpenAI, 2023).

**Evaluation Metrics:** For detection, we report image-level accuracy (ACC) and F1 scores. For localization, we provide Intersection over Union (IoU) and F1 scores. To evaluate interpretability, we use Cosine Semantic Similarity (CSS) to assess the similarity between the predicted text and ground truth text by calculating the cosine similarity between their high-dimensional semantic vectors. For both detection and localization, a default threshold of $0.5$ is applied unless otherwise specified.

**Implementation Details:** On the MMTD-Set, we initially fine-tune the M-LLM using LoRA (rank=128, alpha=256), such as LLaVA-v1.5-13B (Liu et al., 2024), while simultaneously training the Domain Tag Generator with full parameters. The model is trained for $10$ epochs on $4$ NVIDIA A100 40G GPUs, with a learning rate of $2\times10^{-4}$. Afterward, we fine-tune the Tamper Comprehension Module and Segment Anything Model (Kirillov et al., 2023) with LoRA (rank=8, alpha=16), training for $24$ epochs on the same hardware configuration, with a learning rate of $3\times10^{-4}$.

### 4.2 COMPARISON WITH IMAGE FORGERY DETECTION METHOD

To verify the superiority and generalization of our method on the image forgery detection task, we test the detection accuracy on MMTD-Set (Photoshop, DeepFake, AIGC-Editing). As shown in

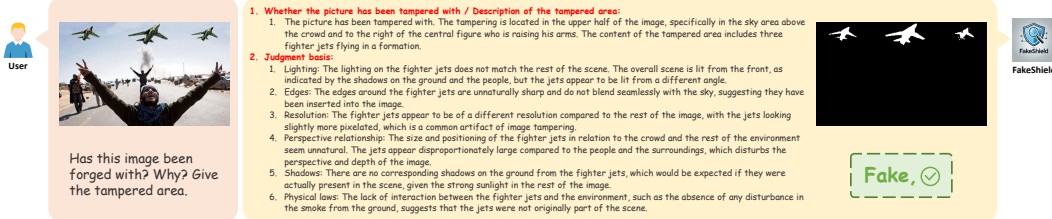

Figure 4: Detection, localization and explanation results of our FakeShield.

Table 1, our FakeShield almost achieves optimal performance across various tampering and testing data domains. For example, our method outperforms the second-best method, MVSS-Net, with an ACC of 0.08 and an F1 of 0.05 on the IMD2020 dataset. Notably, since we introduce the domain-tag guidance strategy, our method not only achieves excellent performance on the traditional IFDL benchmark but also generalizes to DeepFake and AIGC tampering, achieving 0.93 and 0.98 detection accuracy, respectively. However, other works lack an effective mechanism to handle data domain conflicts. Even when trained using the same multi-data domain training set as ours, their detection accuracy remains insufficient.

Furthermore, we compare our method with some recent DeepFake detection approaches on the DFFD (FaceApp, FFHQ) and Seq-DeepFake, where Exposing and RECCE were retrained on our datasets. As reported in Table 2, our approach significantly outperforms all other methods. Notably, the domain tag mechanism not only alleviates conflicts across diverse data domains but also promotes complementarity and mutual enhancement between them. Although specifically designed for DeepFake detection, these methods still underperform compared to our FakeShield.

Table 2: Performance comparison on DFFD and Seq-DeepFake datasets.

| Method | DFFD | | Seq-DeepFake | |
|---|---|---|---|---|
| | ACC | F1 | ACC | F1 |
| CADDM | 0.52 | 0.60 | 0.53 | 0.59 |
| HiFi-DeepFake | 0.52 | 0.64 | 0.52 | 0.57 |
| Exposing | 0.82 | 0.84 | 0.71 | 0.78 |
| RECCE | 0.92 | 0.92 | 0.75 | 0.79 |
| **FakeShield** | **0.98** | **0.99** | **0.84** | **0.91** |

### 4.3 COMPARISON WITH M-LLMS

To assess the quality of explanation text generation, we employ the tampered descriptions generated by GPT-4o as ground truth on the MMTD-Set(Photoshop, DeepFake, AIGC-Editing), using cosine semantic similarity (CSS) to compare the performance of pre-trained M-LLMs against FakeShield. The results in Table 3 show that our approach consistently achieves the best performance across nearly all test sets. For instance, on the DSO, our method attains a CSS score of 0.8873, significantly surpassing the second-best result from InternVL2-26B. Figure 4 presents an example of our explanation. The model identifies the three airplanes in the sky as fake and explains its judgment based on lighting and edges, providing an accurate manipulation region mask.

Notably, some pre-trained M-LLMs exhibit limited proficiency in detecting tampered content. For instance, when tampering causes clear physical law violations, these M-LLMs leverage their pre-training knowledge to make reasonably correct judgments. However, they struggle to perform more precise analyses, like detecting lighting or perspective inconsistencies, reducing overall accuracy.

Table 3: Comparative results(CSS↑) of the pre-trained M-LLMs and FakeShield in tampering explanation capabilities on the MMTD-Set.

| Method | PhotoShop | | | | | | | DeepFake | AIGC-Editing |
|---|---|---|---|---|---|---|---|---|---|
| | CASIA1+ | IMD2020 | Columbia | Coverage | NIST | DSO | Korus | | |
| GPT-4o | 0.5183 | 0.5326 | 0.5623 | 0.5518 | 0.5732 | 0.5804 | 0.5549 | 0.5643 | 0.6289 |
| LLaVA-v1.6-34B | 0.6457 | 0.5193 | 0.5578 | 0.5655 | 0.5757 | 0.5034 | 0.5387 | 0.6273 | 0.6352 |
| InternVL2-26B | 0.6760 | 0.5750 | 0.6155 | 0.6193 | 0.6458 | 0.6484 | 0.6297 | 0.6570 | 0.6751 |
| Qwen2-VL-7B | 0.6133 | 0.5351 | 0.5603 | 0.5702 | 0.5559 | 0.4887 | 0.5260 | 0.6060 | 0.6209 |
| FakeShield | **0.8758** | **0.7537** | **0.8791** | **0.8684** | **0.8087** | **0.8873** | **0.7941** | **0.8446** | **0.8860** |

### 4.4 COMPARISON WITH IMAGE FORGERY LOCATION METHOD

To assess the model's capability to locate tampered regions, we conduct comparisons with some competitive IFDL methods on the MMTD-Set. As present in Table 4, our method consistently

Table 4: Comparative results of tamper localization capabilities between competing IFDL methods and FakeShield, tested on MMTD-Set(Photoshop, DeepFake, AIGC-Editing).

| Method | CASIA1+ | | IMD2020 | | Columbia | | NIST | | DSO | | Korus | | DeepFake | | AIGC-Editing | |
|---|---|---|---|---|---|---|---|---|---|---|---|---|---|---|---|---|
| | IoU | F1 | IoU | F1 | IoU | F1 | IoU | F1 | IoU | F1 | IoU | F1 | IoU | F1 | IoU | F1 |
| SPAN | 0.11 | 0.14 | 0.09 | 0.14 | 0.14 | 0.20 | 0.16 | 0.21 | 0.14 | 0.24 | 0.06 | 0.10 | 0.04 | 0.06 | 0.09 | 0.12 |
| ManTraNet | 0.09 | 0.13 | 0.10 | 0.16 | 0.04 | 0.07 | 0.14 | 0.20 | 0.08 | 0.13 | 0.02 | 0.05 | 0.03 | 0.05 | 0.07 | 0.12 |
| OSN | 0.47 | 0.51 | 0.38 | 0.47 | 0.58 | 0.69 | 0.25 | 0.33 | 0.32 | 0.45 | 0.14 | 0.19 | 0.11 | 0.13 | 0.07 | 0.09 |
| HiFi-Net | 0.13 | 0.18 | 0.09 | 0.14 | 0.06 | 0.11 | 0.09 | 0.13 | 0.18 | 0.29 | 0.01 | 0.02 | 0.07 | 0.11 | 0.13 | 0.22 |
| PSCC-Net | 0.36 | 0.46 | 0.22 | 0.32 | 0.64 | 0.74 | 0.18 | 0.26 | 0.22 | 0.33 | 0.15 | **0.22** | 0.12 | 0.18 | 0.10 | 0.15 |
| CAT-Net | 0.44 | 0.51 | 0.14 | 0.19 | 0.08 | 0.13 | 0.14 | 0.19 | 0.06 | 0.10 | 0.04 | 0.06 | 0.10 | 0.15 | 0.03 | 0.05 |
| MVSS-Net | 0.40 | 0.48 | 0.23 | 0.31 | 0.48 | 0.61 | 0.24 | 0.29 | 0.23 | 0.34 | 0.12 | 0.17 | 0.10 | 0.09 | **0.18** | **0.24** |
| FakeShield | **0.54** | **0.60** | **0.50** | **0.57** | **0.67** | **0.75** | **0.32** | **0.37** | **0.48** | **0.52** | **0.17** | 0.20 | **0.14** | **0.22** | **0.18** | **0.24** |

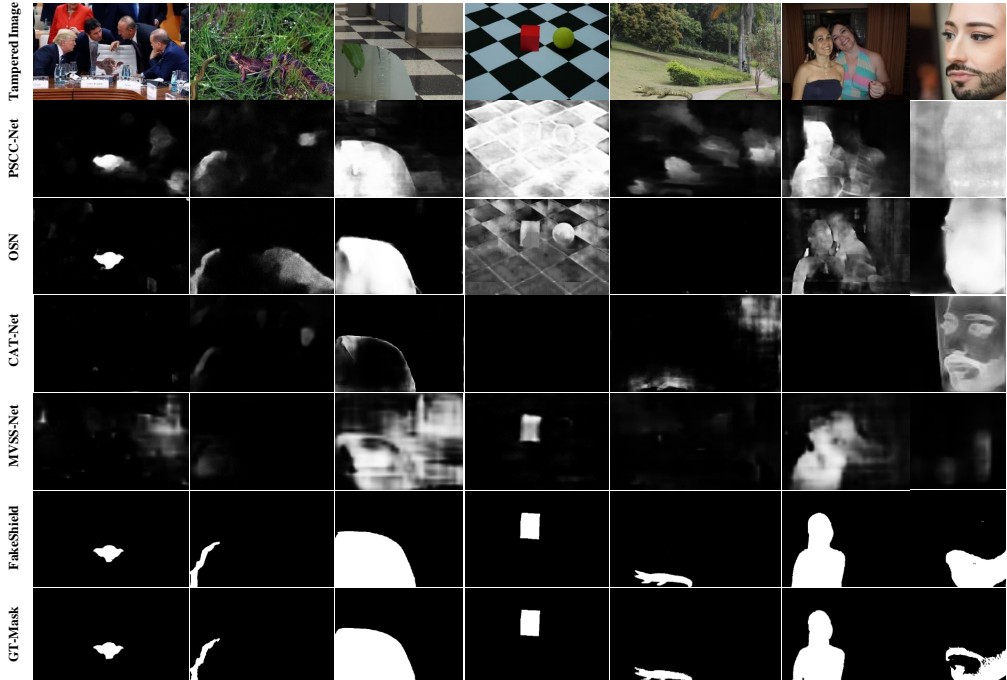

Figure 5: Comparisons between our FakeShield and other competitive methods. The samples, from left to right, are drawn from IMD2020, CASIA1+, Columbia, NIST16, Korus, DSO, and DeepFake.

surpasses other methods across almost all test datasets. For instance, on the IMD2020, our method outperforms the suboptimal method OSN with notable advantages of 0.12 in IoU and 0.1 in F1 score. On the CASIA1+, we also lead OSN with an IoU of 0.07 and an F1 of 0.09.

Additionally, subjective comparisons of several methods are shown in Figure 5, where our approach precisely captures the tampered areas, producing clean and complete masks. In contrast, methods like PSCC-Net exhibit dispersed attention over the image, resulting in blurred segmentations and an overly broad predicted tampering area. Notably, as our segmentation module MFLM is based on the pre-trained visual model SAM, it inherits SAM's powerful semantic segmentation capabilities and can accurately segment targets with clear semantic information (Columns 1 and 2 of Figure 5). Additionally, due to the diverse tampering types in MMTD-Set, our method can also accurately segmenting tampered areas that lack distinct semantic information (Columns 3 of Figure 5).

## 4.5 ROBUSTNESS STUDY

With the widespread use of the Internet and social media, individuals are increasingly receiving images degraded by transmission artifacts such as JPEG compression and Gaussian noise. Our model's performance on MMTD-Set(CASIA1+) under these degradations is reported in Table 5, which includes four common degradation types: JPEG compression qualities of 70 and 80, and Gaussian

noise variances of 5 and 10. As M-LLMs primarily emphasize high-level semantic information, although we do not specifically add degraded data during training, FakeShield demonstrates robustness to low-level visual distortions and noise. JPEG compression and Gaussian noise, commonly associated with social media, have minimal effect on its performance, highlighting the stability, robustness, and practical advantages of our approach.

Table 5: Explanation and location performance under different degradations.

| Method | Explanation | Location | |
|---|---|---|---|
| | CSS | IoU | F1 |
| JPEG 70 | 0.8355 | 0.5022 | 0.5645 |
| JPEG 80 | 0.8511 | 0.5026 | 0.5647 |
| Gaussian 5 | 0.8283 | 0.4861 | 0.5494 |
| Gaussian 10 | 0.8293 | 0.4693 | 0.5297 |
| Original | 0.8758 | 0.5432 | 0.6032 |

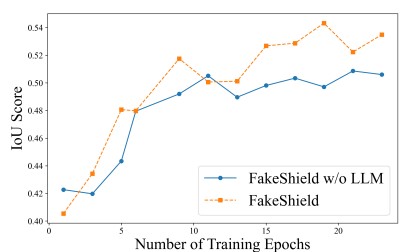

Figure 6: Ablation study on the DTE-FDM.

## 4.6 ABLATION STUDY

**Ablation Study on Domain Tag Generator:** To validate that the proposed domain tag effectively mitigates domain discrepancies and enhances the model's generalization across diverse data domains, we conducted an ablation study. Specifically, we removed the domain tag generator (DTG) and trained the FakeShield with identical configurations on the MMTD-Set, the test results are displayed in Table 6. Without the DTG, the model's detection performance declined across test sets from each data domain. Notably, the detection ACC and F1 score for IMD2020 decreased by 0.12 and 0.11. This demonstrates that without the support of the DTG module, the model struggles to effectively differentiate between various data domains, leading to more pronounced data conflicts and a significant reduction in both generalization and practical applicability.

**Ablation Study on LLM in the DTE-FDM:** Furthermore, to verify the necessity of the LLM in the DTE-FDM, we design a variant of FakeShield, which removes the LLM and directly input $\{\mathbf{T}_{ins}, \mathbf{T}_{tag}, \mathbf{T}_{img}\}$ into the tamper comprehension module, adjusting its training objective to directly produce the description

Table 6: Performance comparison of FakeShield with and without DTG on different datasets.

| Method | CASIA1+ | | IMD2020 | | DeepFake | | AIGC-Editing | |
|---|---|---|---|---|---|---|---|---|
| | ACC | F1 | ACC | F1 | ACC | F1 | ACC | F1 |
| Ours w/o DTG | 0.92 | 0.92 | 0.71 | 0.79 | 0.89 | 0.90 | 0.72 | 0.78 |
| Ours | **0.95** | **0.95** | **0.83** | **0.90** | **0.98** | **0.99** | **0.93** | **0.93** |

$\mathbf{O}_{det}$ and the mask $\mathbf{M}_{loc}$. Using the same training configurations, we train the variant and our original framework for 25 epochs, evaluating their localization accuracy on the CASIA1+ dataset. As shown in Figure 6, after removing the LLM, the localization performance consistently lags behind the original framework throughout the entire training process and it converges earlier. It proves that joint training detection and localization via a single MFLM tends to cause notable performance degradation than our decoupled module design, which further highlights the critical role of our LLM module in enhancing the semantic understanding of the proposed framework.

## 5 CONCLUSION

In this work, we present the first application of an M-LLM for the explanation IFDL, marking a significant advancement in the field. Our proposed framework, FakeShield, excels in tampering detection while delivering comprehensive explanations and precise localization, demonstrating strong generalization across a wide range of manipulation types. These features make it a versatile and practical tool for diverse real-world applications. Looking to the future, this work can play a crucial role in several areas, such as aiding in the improvement of laws and regulations related to digital content manipulation, informing the development of guidelines for generative artificial intelligence, and promoting a clearer and more trustworthy online environment. Additionally, FakeShield can assist in evidence collection for legal proceedings and help correct misinformation in public discourse, ultimately contributing to the integrity and reliability of digital media.

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

# A APPENDIX

## A.1 LIMITATIONS AND FUTURE WORKS

One limitation of our current framework is its suboptimal performance when handling more complex types of deepfake tampering, such as identity switching and full-face generation. These types of manipulations introduce unique challenges that our model currently struggles to address effectively. To address this, future work will focus on several key optimizations. First, we plan to incorporate a Chain-of-Thought (CoT) (Wei et al., 2022) mechanism to enhance the model's reasoning abilities, enabling it to detect more subtle manipulations in deepfake content. Second, we will expand our training dataset to include a broader range of deepfake samples, encompassing various tampering techniques and scenarios, to improve the model's generalization. Finally, we will optimize specific modules within the framework to better handle these complex tampering types, creating a more robust and adaptable detection system. These improvements are expected to significantly enhance the framework's performance across a wider spectrum of deepfake domains.

## A.2 DATA SOURCES

We collected source data from the dataset mentioned in Section 4.1, with the details provided in Table 7.

It is noted that the FaceApp and FFHQ datasets are part of the DFFD (Dang et al., 2020) dataset. We follow their original configuration to divide the training and validation sets. FFHQ (NVIDIA Corporation) consists of real face images, while FaceApp (FaceApp Limited, 2017) contains fake faces generated by the FaceApp (FaceApp Limited, 2017) tool, which manipulates facial attributes in the images.

Regarding the construction process of the AIGC-Editing dataset, we first collected 20,000 real images from the COCO (Lin et al., 2014) dataset and used the SAM (Kirillov et al., 2023) tool to segment the masks of all targets. We then selected the target mask with the third-largest area in each image and applied the Stable-Diffusion-Inpainting (Lugmayr et al., 2022) method to partially redraw this section. To ensure a balance between positive and negative samples, we further extracted an additional 20,000 images from the COCO dataset, distinct from the previously selected ones.

Table 7: Summary of datasets used for training, and evaluation.

| Dataset | Real | Fake | Copy-Move | Splicing | Removal | DeepFake | AIGC-Edit |
|---------|------|------|-----------|----------|---------|----------|-----------|
| **#Training** | | | | | | | |
| Fantastic Reality | 16,592 | 19,423 | - | 19,423 | - | - | - |
| CASIAv2 | 7,491 | 5,123 | 3,295 | 1,828 | - | - | - |
| FFHQ | 10000 | - | - | - | - | - | - |
| FaceAPP | - | 7,308 | - | - | - | 7,308 | - |
| COCO | 20,000 | - | - | - | - | - | - |
| AIGC-Editing | - | 20,000 | - | - | - | - | 20,000 |
| **#Evaluation** | | | | | | | |
| CASIAv1+ | 800 | 920 | 459 | 461 | - | - | - |
| Columbia | - | 180 | - | 180 | - | - | - |
| Coverage | - | 100 | 100 | - | - | - | - |
| NIST | - | 564 | 68 | 288 | 208 | - | - |
| IMD2020 | 414 | 2,010 | - | 2,010 | - | - | - |
| DSO | - | 100 | - | 100 | - | - | - |
| Korus | - | 220 | - | 220 | - | - | - |
| FFHQ | 1,000 | - | - | - | - | - | - |
| FaceAPP | - | 1,000 | - | - | - | 1,000 | - |
| COCO | 20,000 | - | - | - | - | - | - |
| AIGC-Editing | - | 20,000 | - | - | - | - | 20,000 |

### A.3 More Ablation Experiments

**Exploring the Introduction of an Error-Correction Mechanism in MFLM**: We observe that our MFLM inherently possesses some error-correction capability. This is because, during training, we use potentially inaccurate $\mathbf{O}_{det}$ as input and encourage the model to output $\hat{\mathbf{M}}_{loc}$. Thus, during testing, if $\mathbf{O}_{det}$ contains minor inaccuracies, such as incorrect descriptions of tampered locations, MFLM can correct them. To further explore the impact of incorporating an error correction mechanism during MFLM training on localization accuracy, we conducted an ablation study. During MFLM training, we used the ground truth $\hat{\mathbf{O}}_{det}$ to constrain TCM's output to correct the input $\mathbf{O}_{det}$. The results in the Table 8 indicate that correcting $\mathbf{O}_{det}$ does not improve $\mathbf{M}_{loc}$ predictions, possibly due to interference between mask and text optimization.

Table 8: Exploration on introducing error correction in MFLM on localization performance.

| Method | CASIA1+ | | IMD2020 | | DeepFake | | AIGC-Editing | |
|---|---|---|---|---|---|---|---|---|
| | IoU | F1 | IoU | F1 | IoU | F1 | IoU | F1 |
| Using correct $\mathbf{O}_{det}$ | 0.51 | 0.56 | 0.49 | 0.55 | 0.13 | 0.21 | 0.12 | 0.15 |
| FakeShield | **0.54** | **0.60** | **0.50** | **0.57** | **0.14** | **0.22** | **0.18** | **0.24** |

**More Ablation Study on LLM in the DTE-FDM**: To investigate the roles of $\mathbf{T}_{tag}$, $\mathbf{T}_{img}$, and $\mathbf{T}_{ins}$. When training MFLM, we kept other training configurations unchanged and adjusted the inputs to: $\{\mathbf{T}_{ins}, \mathbf{T}_{img}\}$ and $\{\mathbf{T}_{ins}, \mathbf{T}_{tag}\}$. The test results are shown in Table 9. It is evident that using input $\{\mathbf{T}_{ins}, \mathbf{T}_{img}\}$ is slightly better than $\{\mathbf{T}_{ins}, \mathbf{T}_{tag}\}$, but both are inferior to our method with the input $\{\mathbf{O}_{det}, \mathbf{T}_{img}\}$. This indicates that using an LLM to first analyze the basis for tampering and then guiding the localization achieves better results.

Table 9: Ablation Study on LLM in the DTE-FDM with different input combinations.

| Method | CASIA1+ | | IMD2020 | | DeepFake | | AIGC-Editing | |
|---|---|---|---|---|---|---|---|---|
| | IoU | F1 | IoU | F1 | IoU | F1 | IoU | F1 |
| $\mathbf{T}_{ins}, \mathbf{T}_{img}$ | 0.50 | 0.55 | 0.48 | 0.53 | 0.13 | 0.21 | 0.12 | 0.15 |
| $\mathbf{T}_{ins}, \mathbf{T}_{tag}$ | 0.49 | 0.54 | 0.47 | 0.52 | 0.12 | 0.19 | 0.12 | 0.14 |
| $\mathbf{T}_{ins}, \mathbf{T}_{tag}, \mathbf{T}_{img}$ | 0.51 | 0.55 | 0.48 | 0.54 | 0.13 | 0.2 | 0.11 | 0.14 |
| $\mathbf{O}_{det}, \mathbf{T}_{img}$ (Ours) | **0.54** | **0.60** | **0.50** | **0.57** | **0.14** | **0.22** | **0.18** | **0.24** |

### A.4 More AIGC-Editing generalization experiments

To further verify the model's generalization performance in the AIGC-Editing data domain, we expanded our test set by constructing 2,000 tampered images using controlnet inpainting (Zhang et al., 2023) and SDXL inpainting (Podell et al., 2023), which the model had not encountered before. We selected MVSS-Net, CAT-Net, and HiFi-Net, which performed well in the AIGC-Editing data domain in Tables 1, as comparison methods for testing. The detection and localization test results are presented in Table 10. Our method demonstrates leading performance on unseen datasets and exhibits strong generalization capabilities.

Table 10: Generalized performance comparison on ControlNet and SDXL Inpainting datasets.

| Method | ControlNet Inpainting | | | | SDXL Inpainting | | | |
|---|---|---|---|---|---|---|---|---|
| | ACC | Image-level F1 | IoU | Pixel-level F1 | ACC | Image-level F1 | IoU | Pixel-level F1 |
| MVSS-Net | 0.38 | 0.27 | 0.05 | 0.09 | 0.35 | 0.20 | 0.05 | 0.08 |
| CAT-Net | 0.90 | 0.89 | 0.11 | 0.17 | 0.86 | 0.83 | 0.06 | 0.09 |
| HiFi-Net | 0.49 | 0.66 | 0.17 | 0.28 | 0.44 | 0.61 | 0.02 | 0.03 |
| **Ours** | **0.99** | **0.99** | **0.18** | **0.24** | **0.99** | **0.99** | **0.18** | **0.23** |

## A.5 ANSWER ANALYSIS

Figure 7 presents the adjective and noun word clouds for the ground truth (GT) descriptions in the MMTD-Set and the answer descriptions generated by our FakeShield. It is evident that, through effective fine-tuning, FakeShield can be guided by the dataset to assess both image-level semantic plausibility (e.g., "physical law," "texture") and pixel-level tampering artifacts (e.g., "edge," "resolution") to determine whether an image is real.

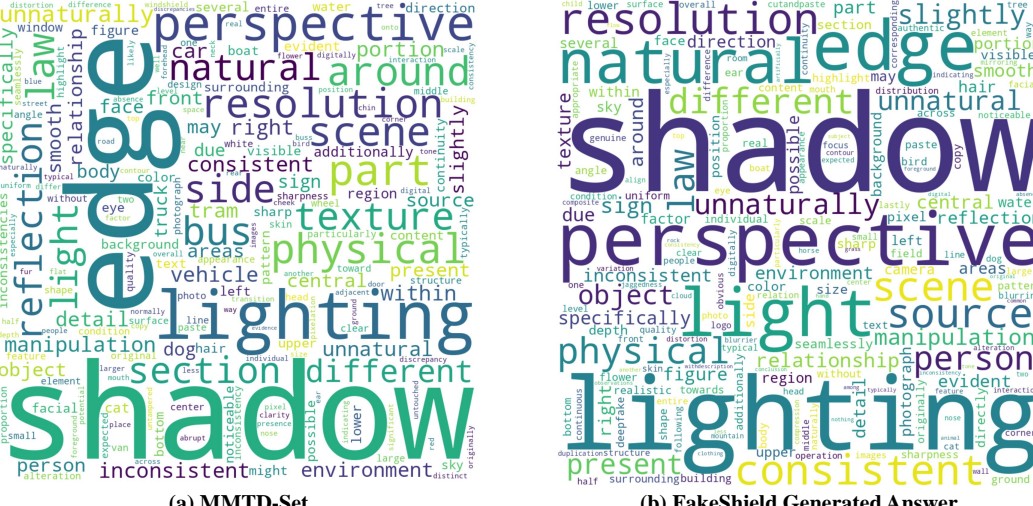

| (a) MMTD-Set | (b) FakeShield Generated Answer |
|---|---|

Figure 7: **The noun and verb word clouds in the MMTD-Set and the FakeShield Generated Answer.** It can be seen that the high-frequency vocabulary of the two is basically the same.

## A.6 PROMPTS

During the process of using GPT-4o to construct the MMTD-Set, we meticulously designed distinct prompts for each category of tampered data to guide GPT-4o in focusing on specific aspects for image analysis, as illustrated in Figure 8 and Figure 9.

## A.7 EXAMPLES

**Comparison of subjective results of mainstream M-LLM**: As mentioned in Section 4.3, we selected some M-LLM output samples, as shown in Figures 10 and 11.

**FakeShield output subjective samples**: We selected several results from FakeShield's testing on PhotoShop, DeepFake, and AIGC-Editing datasets, as displayed in Figures 12, 13, 14, 15, 16 and 17.

**MMTD-Set data set example**: We select some samples from the MMTD-Set data set and display them in Figures 18, 19, and 20.

You are an AI visual assistant that can help humans analyze some tampered images. You will receive two images, the first is the tampered image A, and the second is the binary mask image B of the tampered region (a value of 1 (white) indicates the tampered area, and a value of 0 (black) indicates the untampered area).

Now your task is to use the binary mask provided for the tampered picture A and the tampered area of the picture B:
1. Describe the location of the tampering area in the diagram,
2. Describe in detail the contents of the tampered area,
3. Describe visible details in the image that have been tampered with

In the answer, don't directly mention the binary mask. Always assume that you are simply observing tampered images

When describing Problem 1, use natural language and give both the relative and absolute position of the tampered area, but don't give an ambiguous, unclear description:
(1) Relative position: Use the relative position between objects in the picture, such as above the crowd, on the wall, in the sky, on the table, etc.;
(2) Absolute position: The tampering area is relative to the position of the entire image, such as the left side, right side, top half, bottom half, bottom left corner, lower right corner, upper left corner, upper right corner, etc

When describing Problem 2, please use natural language to describe in detail the types of objects, the number of objects, the actions of objects, and the properties of objects in the tampering area selected by the mask, but do not give ambiguous or unclear descriptions.

When describing Problem 3, consider the visible details caused by tampering from these perspectives, but don't give an ambiguous, unclear description, or something is challenge:
(1) Lighting: Please carefully observe the overall style, color and details of the picture to determine whether there are visual inconsistencies. Pay special attention to the consistency of light, shadows, and colors, and whether there are any unnatural areas or marks. If there are multiple objects or people in the picture, they should be illuminated by the same light and the shadows should be consistent. Inconsistent lighting and shadowing may suggest image compositing or modification.
(2) Copy and paste: Observe whether there are some duplicate areas or blocks in the image, which may be evidence left by the copy and paste operation.
(3) Edges: Check whether there are any unnatural pixel distributions or edges in the image. Particular attention is paid to the presence of discontinuous or inconsistent edges, and the presence of visible shear or synthesis marks.
(4) Resolution: Please check the resolution and compression traces of the picture. Composite or edited images may show unnatural pixel blur, jaggedness, or excessive compression.
(5) Perspective relationship: Observe the perspective and proportion relationship in the picture. Real photos should be consistent in perspective and proportions, and if there are unusual or unnatural perspective relationships, it could be a sign of compositing or editing. Check whether the image has a reasonable change in depth of field, that is, whether the degree of bokeh between the foreground, background, and subject conforms to the actual laws of physics.
(6) Shadows: Observe whether there are reasonable reflections and shadows in the photo. Realistic photos often produce reflections and shadows based on the light source, while composite photos may have unnatural shadows or reflections
(7) Text: If the photo contains words or logos, you can check whether it is clear and legible and consistent with the surrounding environment. Unnaturalness or incongruity of text or logos may suggest tampering with the photograph.
(8) Physical laws: Check whether the content of the picture violates the physical laws, such as the movement trajectory of the object, the position and shape of the reflection, etc.

**(a) Prompt for PhotoShop&AIGC tampered images**

You are an AI visual assistant that helps humans analyze some real images that have not suffered tampering. You will receive an image that is a real image that has not been tampered with

Now your task is to use the provided real image that has not been tampered with A: 1. Describe the angles from which this image can be seen that he has not been tampered with, and can be seen that this is a real and trustworthy real image obtained directly from the camera, 2. Please focus on describing some of the visible details that are consistent with the real scenario, and the real laws of physics, and 3. If you were put in a pile of tampered images, and were asked to judge If you were to judge whether this picture has been tampered with, from what angle would you analyze and judge it and give detailed reasons?

In answering the above questions, consider the visible details triggered by tampering from these perspectives:
(1) Lighting: Please take a close look at the overall style, color and details of the picture to determine whether there is any visual inconsistency. Please pay particular attention to the consistency of lighting, shadows and colors, as well as the presence of unnatural areas or marks. If there are multiple objects or people in the picture, they should be illuminated by the same light and the shadows should be consistent. Inconsistent lighting and shadows may suggest image compositing or modification.
(2) Copy and Paste: Look for signs of duplication or mirroring in the image, which may be evidence left by a copy and paste operation.
(3) Edges: Check for unnatural pixel distribution or edges in the image. Pay particular attention to the presence of discontinuous or inconsistent edges, as well as obvious traces of clipping or compositing.
(4) Resolution: Please check the resolution and traces of compression in the image. Composited or edited images may show unnatural pixel blurring, jaggedness, or excessive compression marks.
(5) Perspective: Observe the perspective and scale relationships in the picture. Real photographs should be consistent in terms of perspective and proportions. If abnormal or unnatural perspective relationships appear, it may be a sign of compositing or editing. Look for reasonable changes in depth of field in the image, i.e., whether the degree of vignetting between the foreground, background and subject is consistent with actual physical laws.
(6) Shadows: Look for reasonable reflections and shadows in the photograph. Real photographs usually produce reflections and shadows accordingly to the light source, while composite photographs may have unnatural shadows or reflections
(7) Text: If a photograph contains text or logos, check that they are legible and in keeping with their surroundings. Unnatural or incongruous text or logos may suggest photo tampering.
(8) Physical laws: Check whether the content of the picture violates physical laws, such as the trajectory of the object's movement, the position and shape of the reflection, and so on.

**(b) Prompt for authentic scene images**

Figure 8: **Prompts for GPT-4o when constructing MMTD-Set.** (a) Analysis guide prompt designed for PhotoShop tampered and AIGC-Editing tampered images. (b) Analysis guidance prompt designed for real scene images.

You are an AI visual assistant that helps humans analyze some images of human faces that have been tampered with by deepfake. You will receive two images, the first is the image A of the face tampered by deepfake and the second is the binary mask image B of the tampered area (a value of 1 (white) indicates the tampered area and a value of 0 (black) indicates the untampered area).

Now, your task is to use the binary masks provided for the tampered regions of the deepfake tampered image A and image B:
1. describe the location of the tampered area in the picture,
2. describe in detail the content of the tampered areas,
3. describes the visible details of the tampered area in the image

In your answer, do not refer directly to the binary mask. Always assume that you are just looking at the tampered image

When describing Problem 1, use natural language and give the location of the tampered area, paying particular attention to the fact that if the tampered area is discontinuous, several larger tampered areas should be described. However, do not give ambiguous and unclear descriptions. Common descriptions of tampered areas include:
Head, Face, Forehead, Eyes, Eyebrows, Eyelids, Eyelashes, Nose, Nostrils, Cheeks, Ears, Mouth, Lips, Chin, Jaw, Hair, Neck, Glasses, Earrings, Headphones, Scalp, Hairline, Sideburns, Temple, Crown, Nape, Jawline, Teeth, Tongue, Gums, Moustache, Beard, Headband, Hat, Tiara, Bandana, Veil, Fascinator, Hairpin, Headscarf

When describing Problem 2, use natural language to describe in detail the types of objects, properties of objects, number of objects, etc. in the tampered area selected by mask, paying special attention to the fact that, if the tampered area is discontinuous, give a description of the tampered areas that are larger in size. But do not give ambiguous or unclear descriptions.

When describing Questions 2 and 3, consider the visible details caused by tampering from these perspectives, but do not give an ambiguous, unclear description that is otherwise challenging:
1. Symmetrical Facial Features: Deepfake-generated faces may exhibit unnaturally perfect symmetry, lacking the subtle asymmetry typically found in real faces.
2. Blur or Distortion Around Edges: Deepfake manipulation may introduce blur or distortion around the edges of the face where the manipulation has taken place, especially if the face has been digitally overlaid onto another body.
3. Inconsistent Lighting and Shadows: Deepfake algorithms may struggle to accurately match the lighting and shadows in the original image, leading to discrepancies in lighting direction or intensity across the face.
4. Unnatural Facial Expressions: Deepfakes may produce facial expressions that appear unnatural, exaggerated, or out of sync with the rest of the image.
5. Mismatched Facial Proportions: Deepfake manipulation may result in facial proportions that are inconsistent with the person's gender or age, such as a man's face on a woman's body or vice versa.
6. Inconsistent Skin Texture and Tone: Deepfake-generated faces may exhibit unnatural skin texture or tone, such as overly smooth or pixelated skin, that differs from the surrounding areas.
7. Missing or Inconsistent Eye Reflections: Deepfake manipulation may result in missing or inconsistent reflections in the eyes, which can provide clues about the authenticity of the image.
8. Change hairstyle: Some deepfake algorithms only tamper with hair, and may change hair color and hairstyle, or add long hair for boys and short hair for girls.
9. Irregularities in Makeup Application: Deepfake manipulation may introduce makeup styles that are inconsistent with the person's gender or age, or exhibit poor application quality.
10. Contextual Inconsistencies: Deepfake-generated images may contain inconsistencies in the overall context of the image, such as discrepancies in perspective, clothing, or surroundings, that suggest manipulation.
11. Unreasonable accessories: Some deepfake algorithms add glasses, earrings, hats, masks, etc. to the image, and the edges, lighting relationships, and perspective of these accessories may be incorrect.
12. If there are glasses or sunglasses in the picture, please pay special attention to whether the glasses or sunglasses have been tampered with or not, the rims, frames, temples, lenses, etc. of the glasses are very susceptible to imperfections and problems.

**(a) Prompt for DeepFake tampered images**

You are an artificial intelligence visual assistant that helps humans analyze some real face images that have not been tampered with by deepfake. You will receive an image of a real face that has not been tampered with by deepfake

Now, your task is to use the provided image of a real face that has not been tampered with by deepfake Answer: 1. Describe the angles from which you can see that this image has not been tampered with by deepfake, and that you can see that this is a real and believable image of a real face straight from the camera, 2. Focus on describing some of the visible details that are consistent with a real face and the real laws of physics, and 3. What would happen if you were put placed in a pile of images that have been tampered with by deepfake and asked to determine whether this image has been tampered with, which perspective would you analyze the judgment from and give detailed reasons?
When describing parts of the human face, common descriptions include
head, face, forehead, eyes, eyebrows, eyelids, eyelashes, nose, nostrils, cheeks, ears, mouth, lips, chin, jaw, hair, neck, eyeglasses, earrings, headphones, scalp, hairline, sideburns, temples, crown, back of the neck, jawline, teeth, tongue, gums, mustache, beard, headband, hat, headdress, headscarf, veil, hairpiece, hairbrush, headband

In answering the above questions, consider the visible details triggered by deepfake tampering from the following perspectives:
1. symmetrical facial features: Deepfake-generated faces may exhibit unnatural perfect symmetry, lacking the subtle asymmetries common in real faces.
2. Blurred or distorted edges: Deepfake processing may result in blurred or distorted edges on processed faces, especially when the face is digitally superimposed on another body.
3. Inconsistent Lighting and Shadows: The Deepfake algorithm may have difficulty accurately matching the lighting and shadows in the original image, resulting in differences in the direction or intensity of lighting across the face.
4. Unnatural facial expressions: Deepfake may produce facial expressions that look unnatural, exaggerated, or out of sync with the rest of the image.
5. Disproportionate Facial Proportions: Deepfake operations may result in facial proportions that do not correspond to the gender or age of the character, such as a man's face on a woman's body, or vice versa.
6. Inconsistent skin texture and tone: Deepfake-generated faces may exhibit unnatural skin texture or tone, such as overly smooth or pixelated skin that is different from the surrounding area.
7. Missing or inconsistent eye reflections: Deepfake operations may result in missing or inconsistent eye reflections, which may provide clues to the realism of the image.
8. Altered hairstyles: Some Deepfake algorithms only tamper with hair, and may change hair color and hairstyle, or add long hair for boys and short hair for girls.
9. Makeup irregularities: Deep forgery algorithms may introduce makeup styles that are inconsistent with an individual's gender or age, or exhibit poor makeup quality.
10. Contextual inconsistencies: Images generated by Deepfake may contain inconsistencies with the overall context of the image, such as differences in perspective, clothing, or surroundings, suggesting manipulation of the image.
11. Unreasonable accessories: Some deepfake algorithms add glasses, earrings, hats, masks, etc. to an image, and the edges, lighting relationships, and perspective of these accessories may be incorrect.

**(b) Prompt for authentic face images**

Figure 9: **More prompts for GPT-4o when constructing MMTD-Set.** (a) Analysis guide prompt designed for DeepFake tampered images. (b) Analysis guidance prompt designed for real face images.

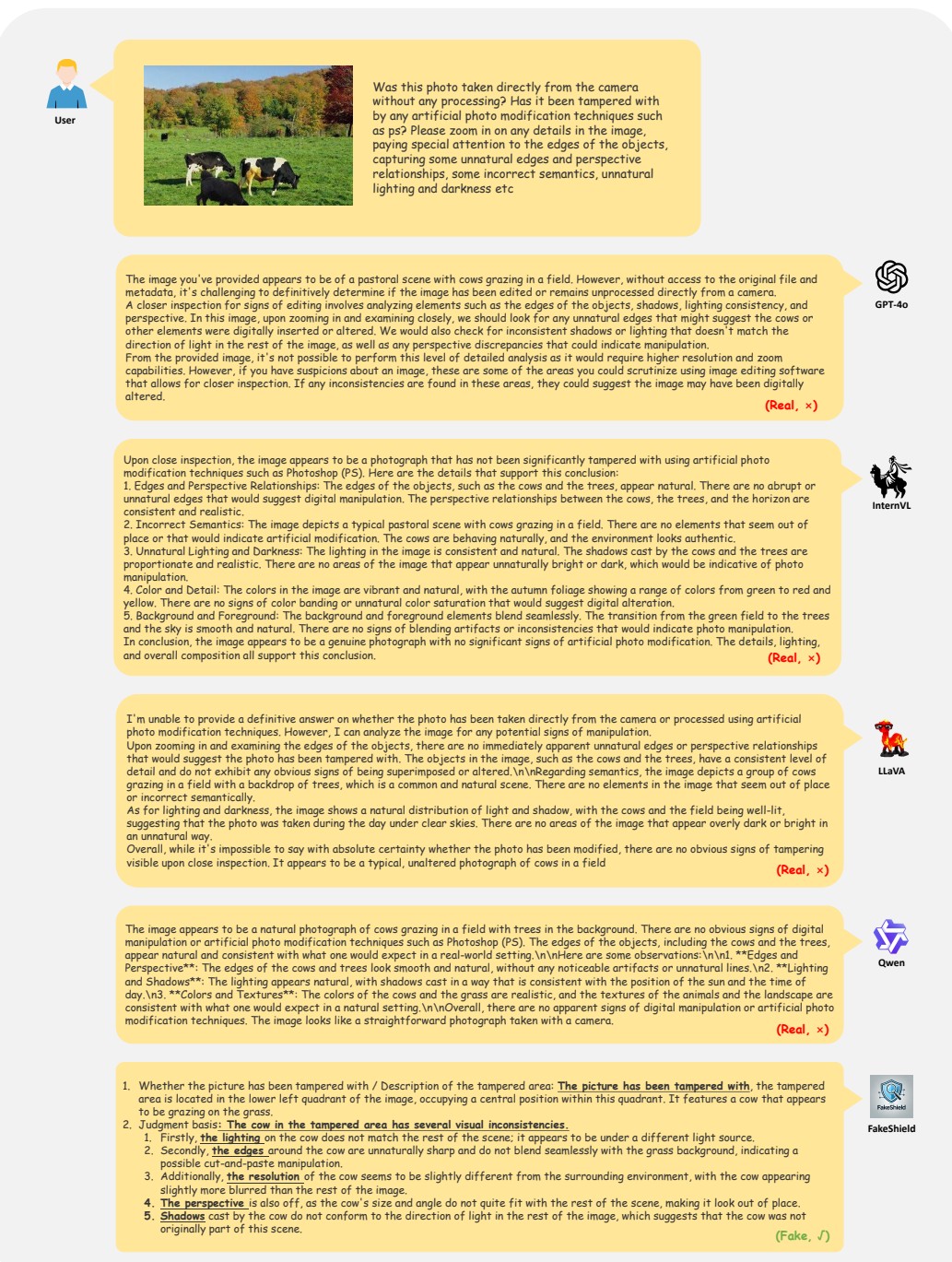

Figure 10: **The response of mainstream pre-trained M-LLM and FakeShield to tampered pictures.** This is a sample from the CASIA1+ (Dong et al., 2013) test set, which has been tampered with by Splicing. The black animal at the bottom of the picture is pasted later. It can be seen that only FakeShield made the correct judgment and accurately described the tampered area.

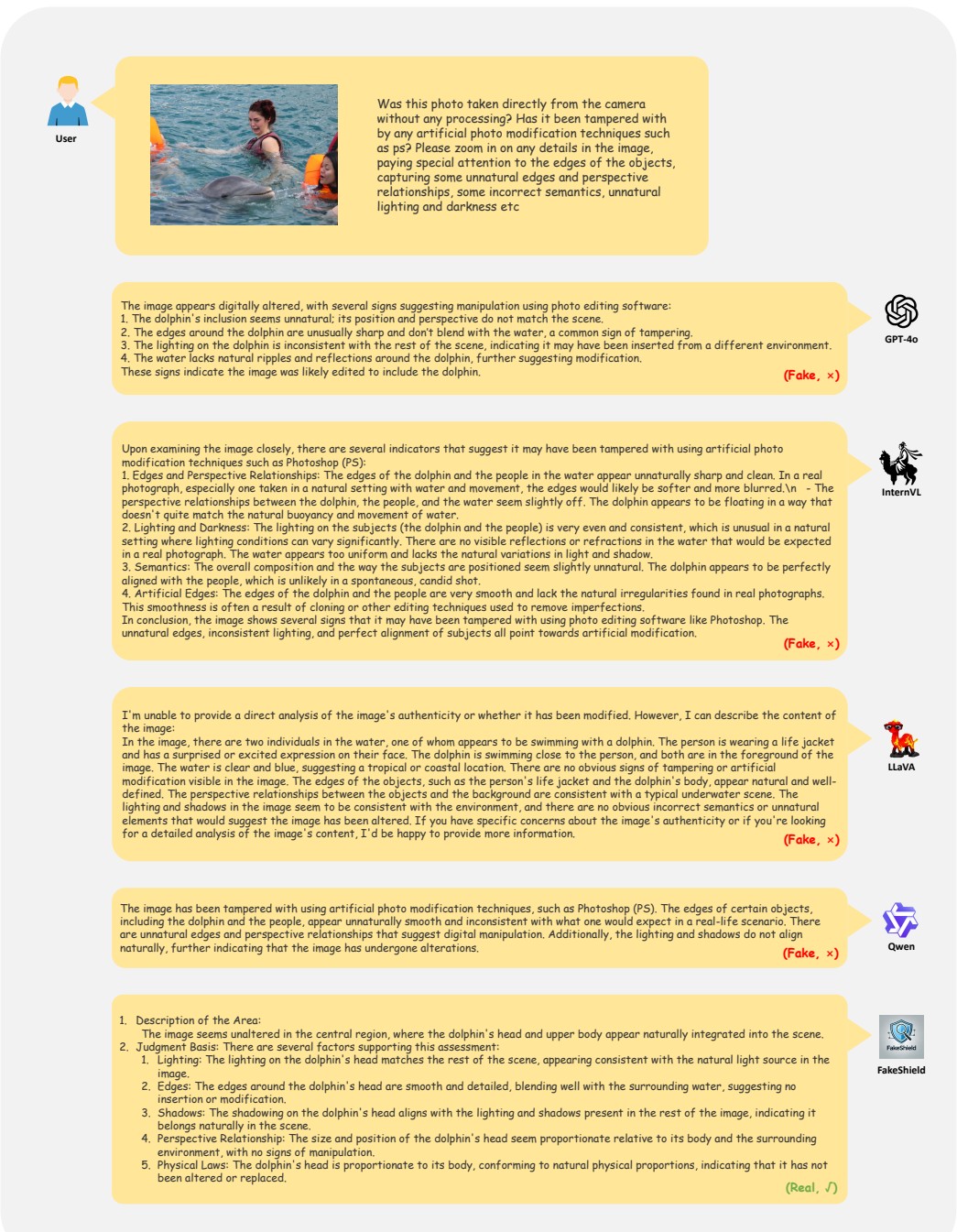

Figure 11: **The response of mainstream pre-trained M-LLM and FakeShield for real images.** This is a sample from the IMD2020 (Novozamsky et al., 2020) test set, which is a real picture. It can be seen that only FakeShield makes the correct judgment, the rest of the methods consider the image to be tampered with.

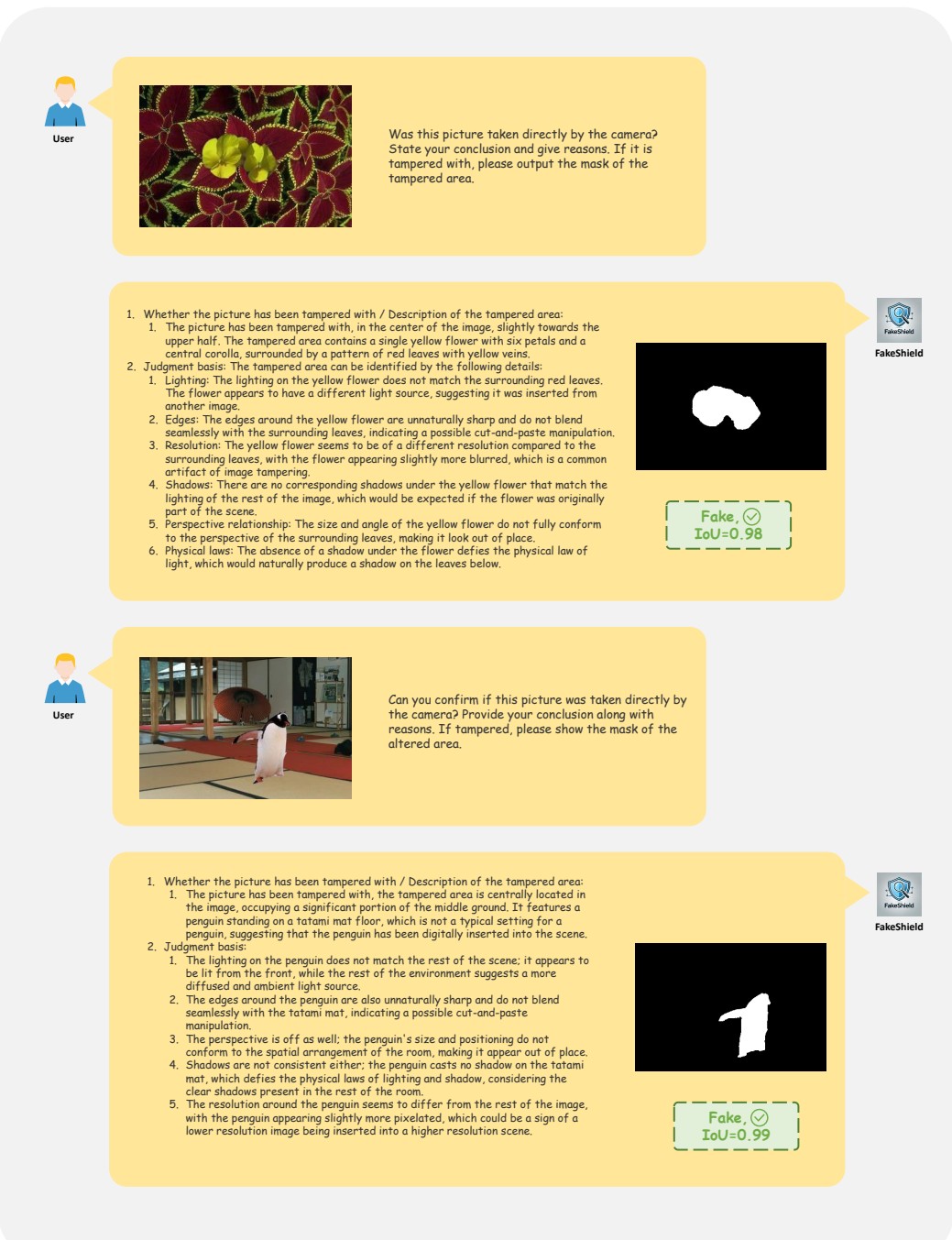

Figure 12: **Dialogue examples of FakeShield when receiving PhotoShop tampered images.** The first image comes from the CASIA1+ (Dong et al., 2013) dataset and has been tampered with by Copy-Move. The yellow petals in the center of the picture are the parts pasted later. The second picture comes from the CASIA1+ (Dong et al., 2013) dataset and has been tampered with by Splicing. The penguin in the center of the picture is the part pasted later.

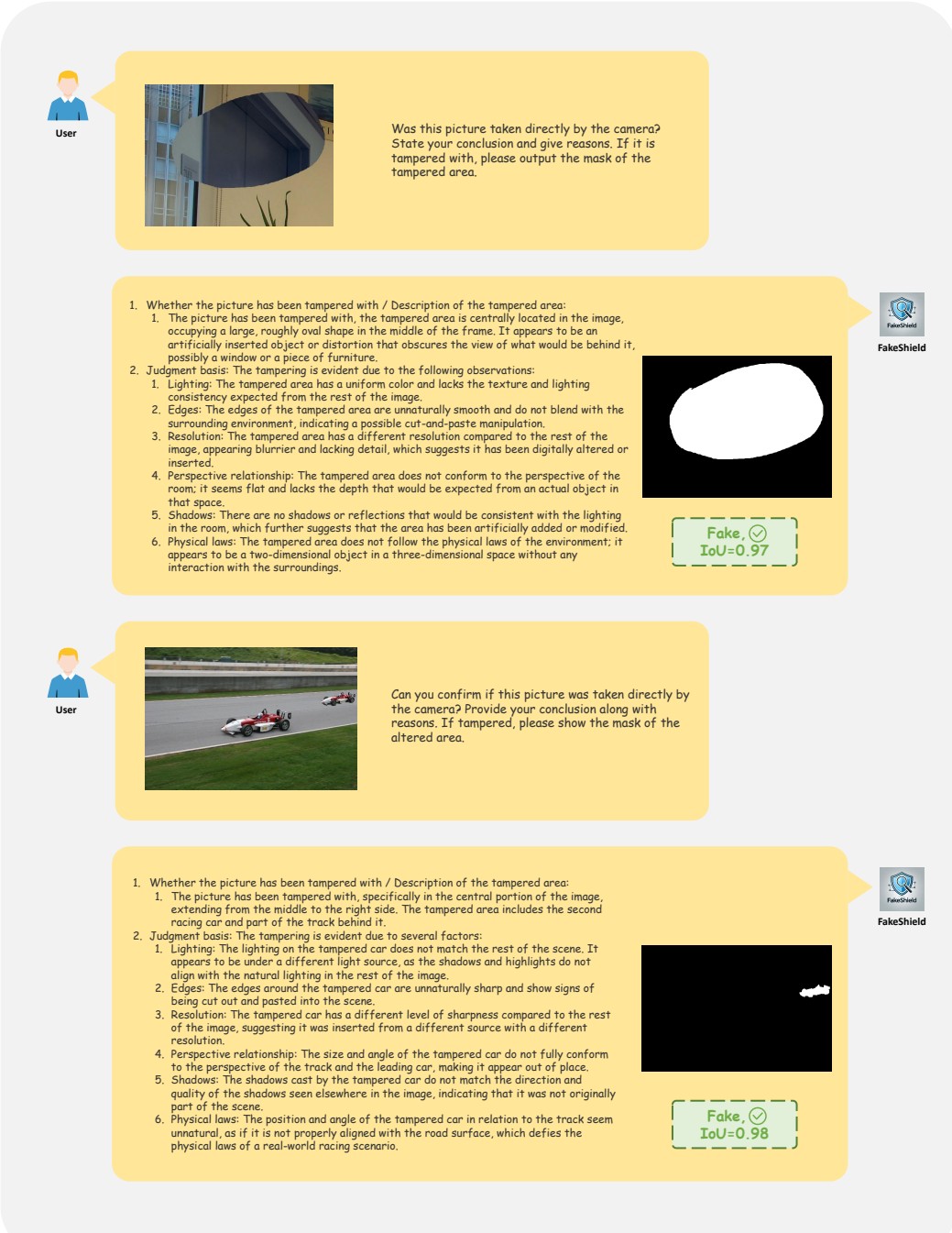

Figure 13: **More dialogue examples of FakeShield when receiving PhotoShop tampered images.** The first image comes from the Columbia (Ng et al., 2009) dataset and has been tampered with by Splicing. The elliptical area in the center of the screen is the post-processing area. The second picture comes from the IMD2020 (Novozamsky et al., 2020) dataset and has been tampered with by Copy-Move. The car on the right side of the image is copied and pasted from the first car.

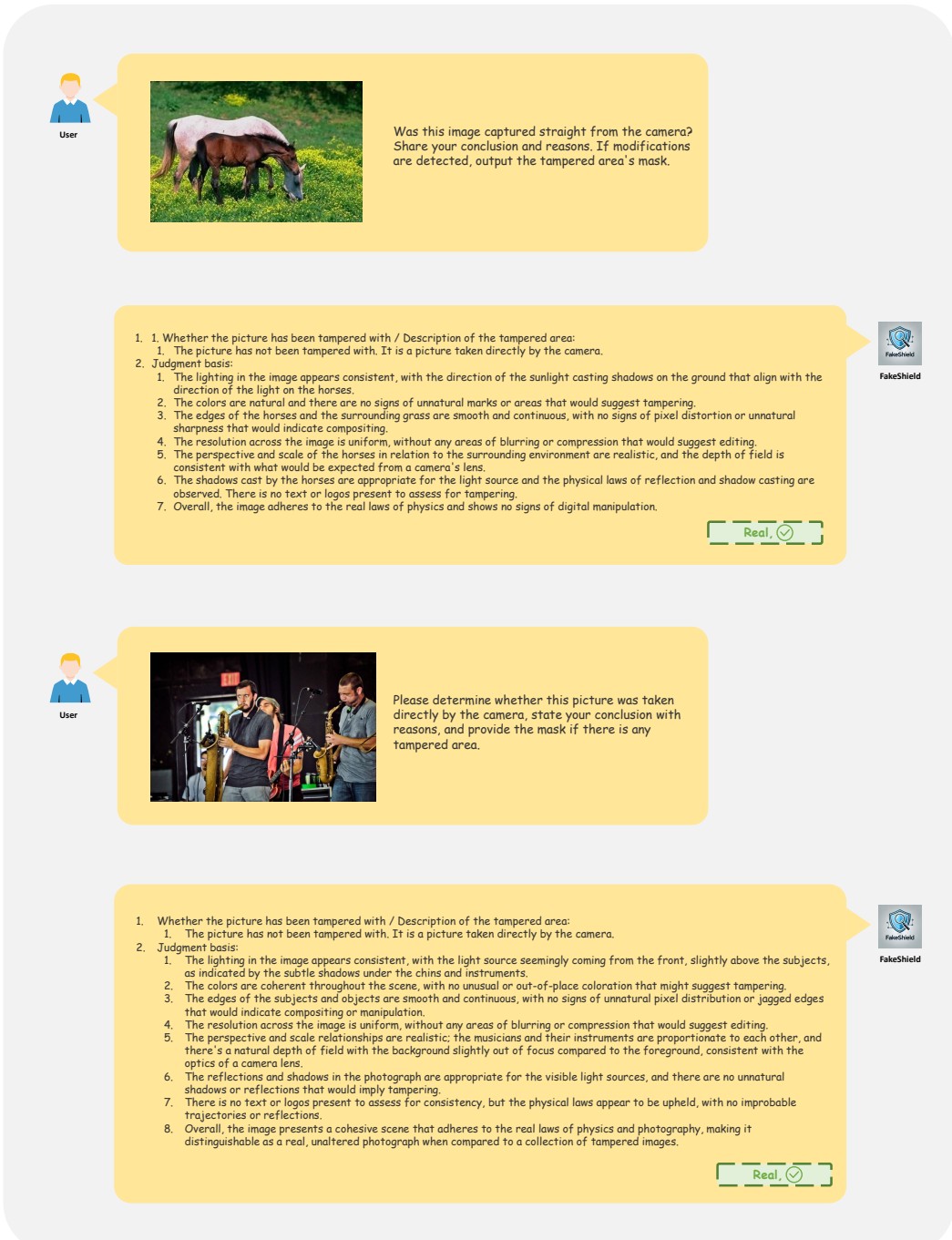

Figure 14: **Dialogue examples of FakeShield when receiving authentic scene images.** The first image comes from the CASIA1+ (Dong et al., 2013) dataset. The second image comes from the IMD2020 (Novozamsky et al., 2020) dataset.

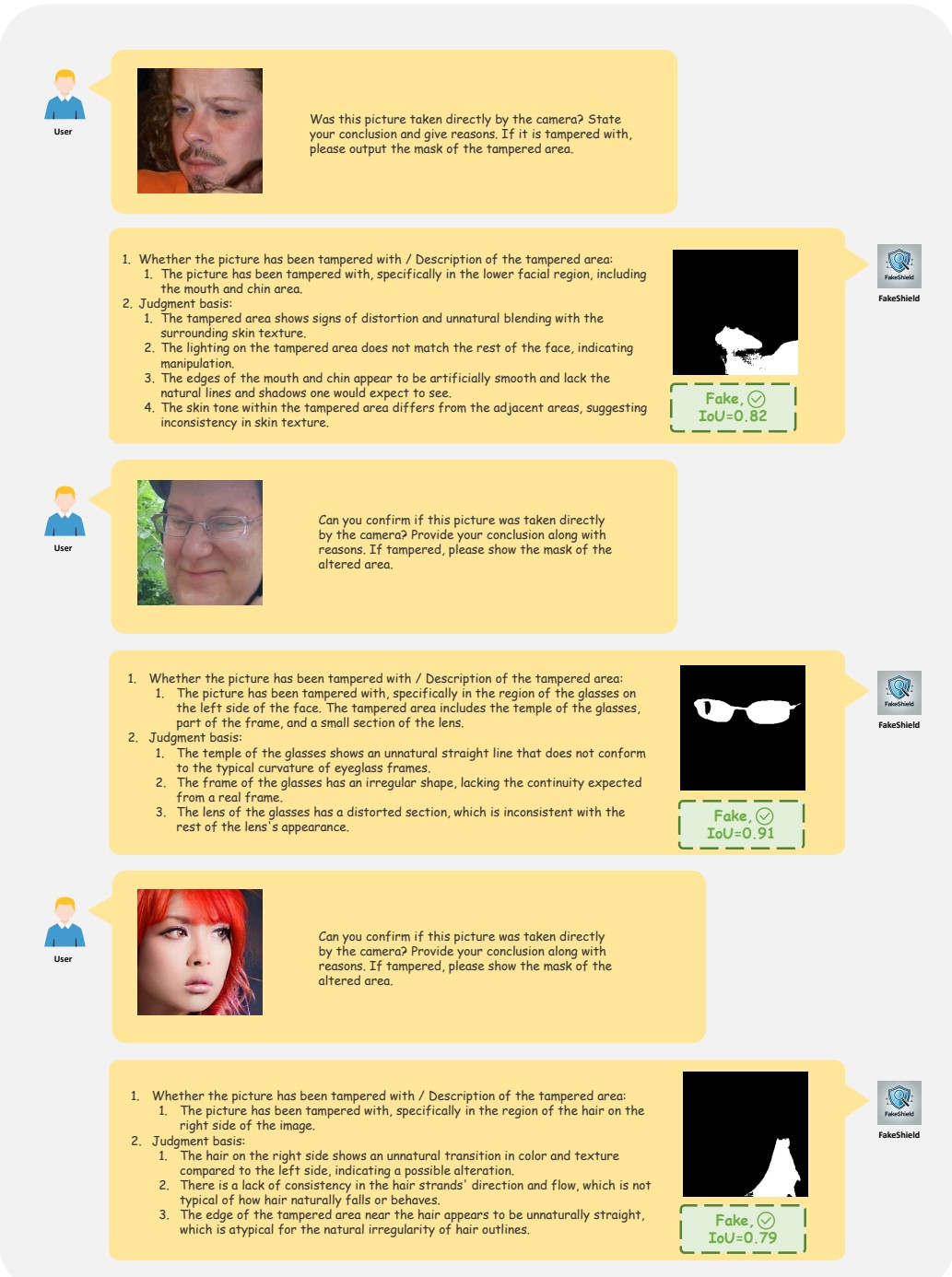

Figure 15: **Dialogue examples of FakeShield when receiving DeepFake tampered images.** The three images all come from the DFFD (Dang et al., 2020) dataset and have been tampered with by FaceAPP (FaceApp Limited, 2017). The first picture has a beard added to the chin, the second picture has glasses added, and the third picture has some hair added to the right side of the face.

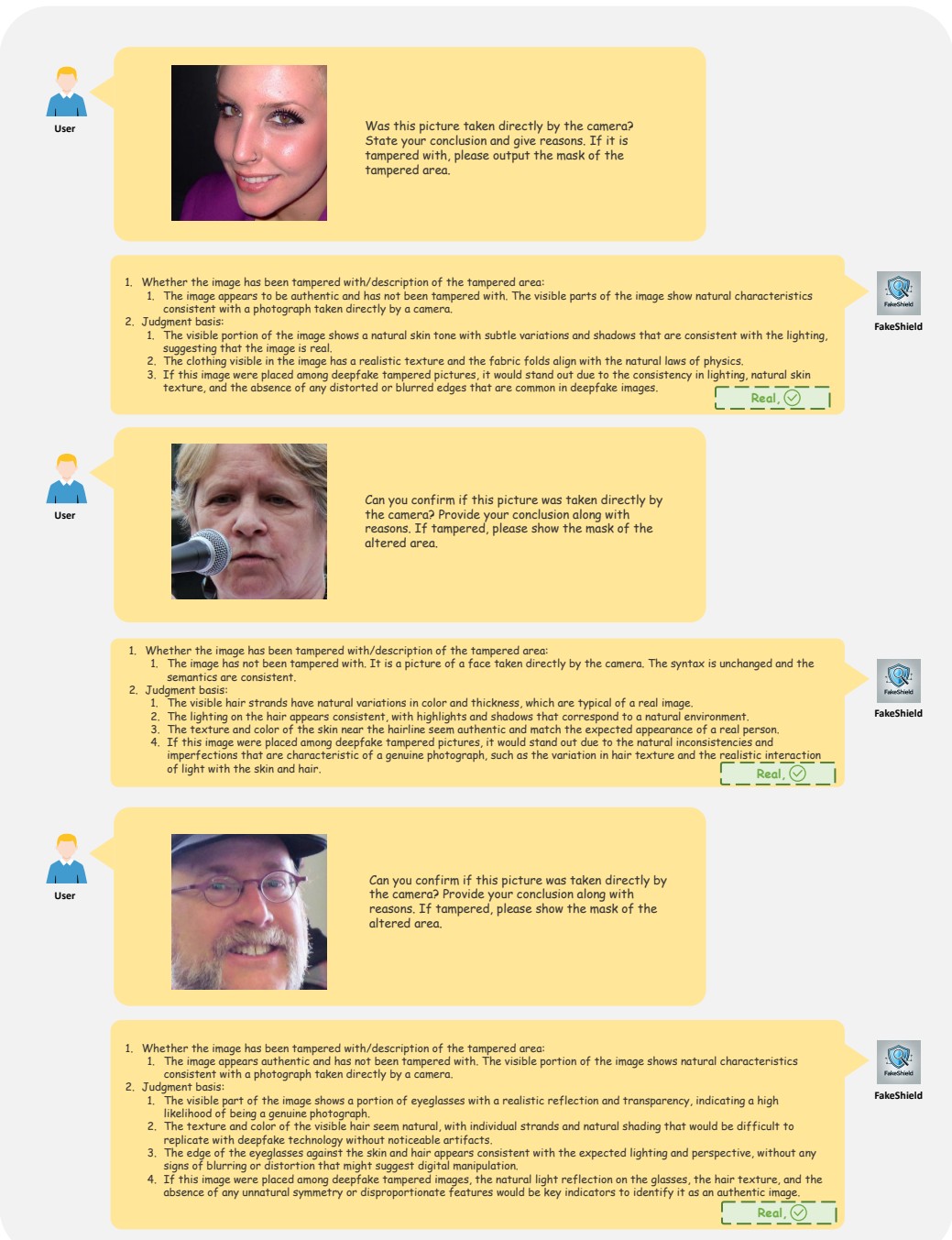

Figure 16: **Dialogue examples of FakeShield when receiving authentic face images.** The three images all come from the DFFD (Dang et al., 2020) dataset.

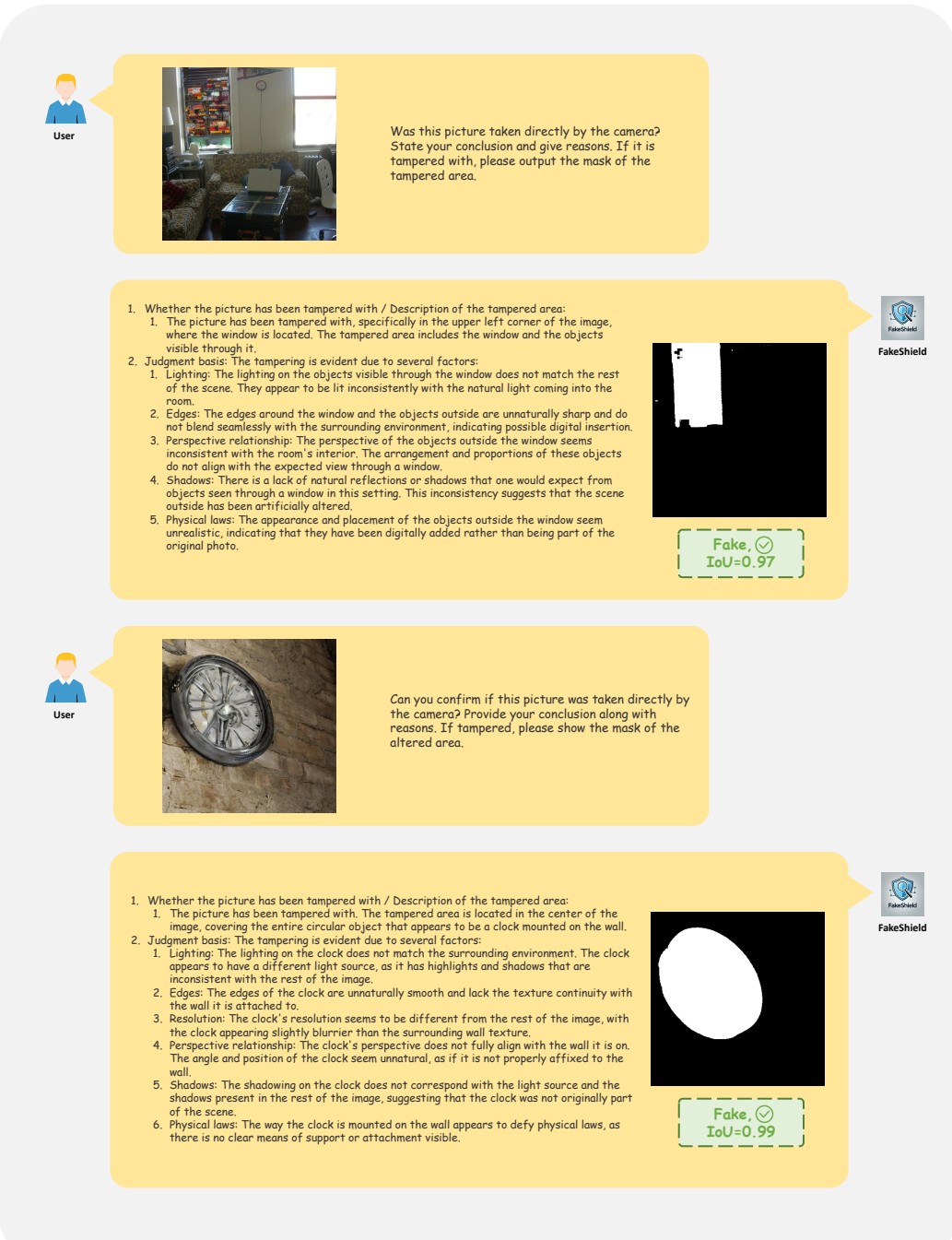

Figure 17: **Dialogue examples of FakeShield when receiving AIGC-Editing tampered images.** The three images all come from the self-generated AIGC-Editing dataset and have been tampered with by Stable-Diffusion-Inpainting (Lugmayr et al., 2022). The window in the upper left corner of the first picture has been redrawn, and the circular clock area on the left side of the second picture has been redrawn.

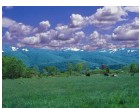

Was this image captured straight from the camera? Share your conclusion and reasons. If modifications are detected, output the tampered area's mask.

1. Whether the picture has been tampered with / Description of the tampered area: The picture has been tampered with, in the upper section, spanning across the entire width of the image. The tampered area includes the mountain range and the sky immediately above it.
2. Judgment basis:
   1. The tampering is evident by the unnaturally smooth transition between the mountain tops and the sky, lacking detailed texture that would be expected in a natural setting.
   2. There is a lack of consistent lighting and shadows on the mountains which do not match with the rest of the image, indicating that this part of the image may have been inserted or altered. The edges of the mountains against the sky are unusually crisp without gradation or atmospheric haze that would normally occur due to distance.
   3. Additionally, the pixelation around the edges of the mountains does not match the resolution of the foreground, suggesting a composite of two different images.

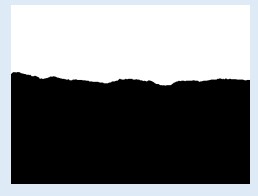

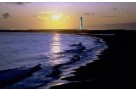

Was this image captured straight from the camera? Share your conclusion and reasons. If modifications are detected, output the tampered area's mask.

1. Whether the picture has been tampered with / Description of the tampered area: The picture has been tampered with, specifically in the area around the lighthouse. The tampered area is located in the upper right corner of the image, against the horizon where the sky meets the sea.
2. Judgment basis: There are several details that suggest tampering:
   1. Lighting: The lighthouse appears to be unnaturally bright compared to the surrounding environment, especially given that the sun is setting and the light is diffused.
   2. Edges: The lighthouse has a somewhat blurry outline that does not match the sharpness of the horizon line, indicating it may have been artificially inserted or altered.
   3. Shadows: The lighthouse does not cast a shadow that is consistent with the setting sun's direction and the existing shadows in the image.
   4. Perspective relationship: The scale of the lighthouse in relation to its placement on the horizon seems slightly off, making it appear as if it does not quite belong there.
   5. Physical laws: The lighthouse's reflection in the water does not match the angle of the sunlight or the other reflections in the water, breaking the consistency expected in a real photo.

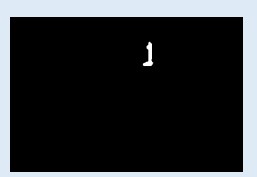

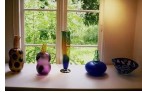

Was this image captured straight from the camera? Share your conclusion and reasons. If modifications are detected, output the tampered area's mask.

1. Whether the picture has been tampered with / Description of the tampered area: The picture has not been tampered with. It is a picture taken directly by the camera.
2. Judgment basis:
   1. Lighting: The lighting in the image appears natural and consistent. The light source seems to be coming from outside the window, casting shadows and highlights appropriately on the window sill and the objects placed on it. The soft diffuse light suggests an overcast day or a shaded area outside, common in real-life scenarios.
   2. Copy and Paste: There is no visible sign of duplication or mirroring that would suggest a copy and paste operation. Each object has unique reflections, colors, and shadows.
   3. Edges: The edges of the objects, window, and foliage outside are smooth and continuous, with no visible signs of pixel distortion or abrupt changes that would indicate tampering.
   4. Resolution: The resolution of the image appears uniform, with no areas showing signs of blurring or jaggedness that would suggest a composite image.
   5. Perspective: The perspective and scale of the objects on the window sill are consistent with how they would appear in real life. The size of the objects relative to each other and their placement gives a sense of depth that aligns with normal visual perception.

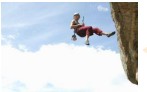

Was this image captured straight from the camera? Share your conclusion and reasons. If modifications are detected, output the tampered area's mask.

1. Whether the picture has been tampered with / Description of the tampered area: The image appears not to have been tampered with. It seems to be a picture taken directly by the camera.
2. Judgment basis:
   1. Lighting: The lighting on the climber is consistent with the direction of sunlight, casting a natural shadow on the climber's body, which conforms to the direction and quality of the lighting in the environment.
   2. Copy and Paste: There are no signs of duplication or mirroring that would indicate a copy and paste operation. \n- Edges: The edges of the climber and the rock face appear natural with no signs of unnatural pixel distribution or obvious traces of clipping.
   3. Resolution: The image maintains a consistent resolution throughout, with no unnatural blurring or jaggedness that would suggest tampering.
   4. Perspective: The scale and perspective of the climber against the rock face are realistic, and the proportions seem accurate.
   5. Shadows: The shadows cast by the climber are consistent with the light source coming from the upper left, and there are no unnatural shadows that would imply manipulation.
   6. Text: There is no visible text or logos to assess for consistency.
   7. Physical laws: The climber's position and equipment behave as expected under the force of gravity, and there are no violations of physical laws.

Figure 18: MMTD-Set data samples related to PhotoShop tampering. The four images are all from the CASIAv2 data set. The first two pictures have been tampered with, but the last two pictures have not been tampered with.

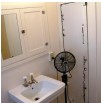

Was this image captured straight from the camera? Share your conclusion and reasons. If modifications are detected, output the tampered area's mask.

1. Whether the picture has been tampered with / Description of the tampered area:
   1. The picture has been tampered with, the tampered area is on the right side of the image, extending vertically from the top to the bottom of the frame. It includes the edge of a bathroom wall and a portion of a black pedestal fan that stands next to the sink.
2. Judgment basis: The tampered area shows several inconsistencies.
   1. The edge of the wall on the right side has a jagged, unnatural outline that does not align with the typical straight lines found in interior architecture.
   2. Moreover, the pedestal fan has a distorted shape, with the top part of the fan appearing disconnected and floating without support.
   3. The shadows and lighting in the tampered area do not match the rest of the image; the shadow on the tampered part of the fan is inconsistent with the lighting on the rest of the fan and the surrounding environment.
   4. There are no clear reflections or shadows corresponding to the tampered section of the wall, suggesting an unnatural alteration. The resolution in the tampered area is also inconsistent, with some parts appearing blurrier than the rest of the image.
   5. These signs indicate that the image has been edited."

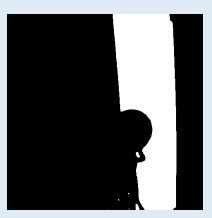

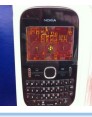

Was this image captured straight from the camera? Share your conclusion and reasons. If modifications are detected, output the tampered area's mask.

1. Whether the picture has been tampered with / Description of the tampered area: The picture has been tampered with. The tampering is located on the screen of the Nokia phone, occupying almost the entire screen area, with only a small untampered margin visible around the edges.
2. Judgment basis:
   1. The contents of the tampered area include jumbled and inverted text, numbers, and symbols that do not conform to the typical display of a phone screen.
   2. The colors within the tampered area are inconsistent with the original screen display, showing a mix of orange, yellow, and white that appear unnatural.
   3. The text and numbers are also not aligned or positioned as expected for a phone's user interface, suggesting a manipulation of the original content. Additionally, there is a lack of continuity with the phone's expected graphical interface, which would normally display information in an organized and legible manner."

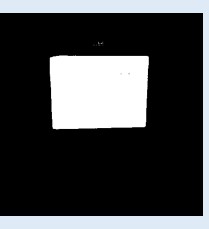

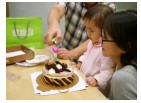

Was this image captured straight from the camera? Share your conclusion and reasons. If modifications are detected, output the tampered area's mask.

1. Whether the picture has been tampered with / Description of the tampered area: The picture has not been tampered with. It appears to be a genuine photograph captured directly by a camera.
2. Judgment basis:
   1. The lighting in the image is consistent, with the shadows on the cake and the hands indicating a light source coming from the upper left side, which aligns with the natural shadow casting.
   2. There are no obvious signs of duplication or mirroring, and the edges of the subjects and objects are natural without any discontinuities or irregular pixel distributions.
   3. The resolution and quality of the image are uniform, with no areas showing signs of blurring or excessive compression that would suggest manipulation. The perspective and scale of the objects and people are realistic and proportionate, and the depth of field is consistent, with a natural focus drop-off towards the back of the image.
   4. The reflections and shadows present in the image behave as expected under the lighting conditions. Text and logos visible in the background are consistent with their surroundings. All content in the picture adheres to physical laws without any anomalies."

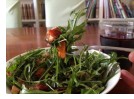

Was this image captured straight from the camera? Share your conclusion and reasons. If modifications are detected, output the tampered area's mask.

1. Whether the picture has been tampered with / Description of the tampered area: The image appears not to have been tampered with. It looks like a photograph taken directly by a camera.
2. Judgment basis: The following details support the conclusion that this is a real, untampered photograph:
   1. Lighting: The lighting on the salad and the glass of liquid appears consistent, with soft, diffused light coming from one general direction, which is plausible for an indoor setting. There are no harsh shadows or inconsistent light spots that would suggest tampering.
   2. Copy and Paste: There are no obvious repeating patterns or mirrored sections that would indicate a copy and paste operation has been conducted.
   3. Edges: The edges of the salad leaves, fork, and other elements in the image look natural with no visible signs of harsh or artificial edges that would suggest cutting or compositing.\n\n- Resolution: The image resolution seems consistent throughout the photograph, with no areas of unnatural blurring or pixelation that would indicate tampering.
   4. Perspective: The perspective and scale of the objects, such as the fork, salad, and glass, are consistent with what would be expected in a real-life setting.
   5. Shadows: The shadows cast by the fork and the leaves in the salad are subtle and match the lighting of the scene. There are no unnatural shadows that would suggest manipulation.

Figure 19: MMTD-Set data samples related to AIGC-Editing tampering. The four images are all from the self-generated AIGC-Editing dataset. The first two pictures have been tampered with, but the last two pictures have not been tampered with.

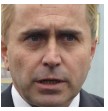

Was this image captured straight from the camera? Share your conclusion and reasons. If modifications are detected, output the tampered area's mask.

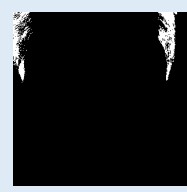

1. Whether the picture has been tampered with / Description of the tampered area: The picture has been tampered with, specifically the hair and ears. The tampered areas include the top and sides of the head, extending to cover the full hair region and the ears on both sides.
2. Judgment basis:
   1. The hair appears unnatural and lacks consistent texture when compared to typical hair, which would normally have more variation in color and strand definition.
   2. The edges of the hair near the forehead and temples are too sharp and show no natural hairline; real hair usually has more irregularity at the hairline.
   3. There is a lack of depth and natural shading around the ears, which can be expected if the image was manipulated; ears typically cast subtle shadows and have more defined contours.
   4. The overall hair shape and the way it sits on the head do not conform to natural hair physics, indicating digital alteration."

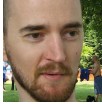

Was this image captured straight from the camera? Share your conclusion and reasons. If modifications are detected, output the tampered area's mask.

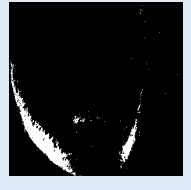

1. Whether the picture has been tampered with / Description of the tampered area:
   1. The picture has been tampered with, specifically in the lower facial region including the jawline and the chin, as well as a small portion of the neck.
2. Judgment basis:
   1. The tampered area appears to have an inconsistent skin tone compared to the surrounding untampered skin, indicating a possible mismatch in texture and color.
   2. The edges of the tampered area around the jawline and chin show signs of being unnaturally smooth, lacking the typical definition and texture found in these facial features.
   3. There is a lack of natural shadowing and lighting in the tampered area, which does not correspond with the lighting present in the rest of the image."

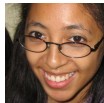

Was this image captured straight from the camera? Share your conclusion and reasons. If modifications are detected, output the tampered area's mask.

1. Whether the image has been tampered with/description of the tampered area: The image does not appear to have been tampered with. The visible parts of the image, including hair and ears, display natural variations and asymmetries typical of a real photograph taken directly by a camera.
2. Judgment basis: The hair shows natural variations in texture and color, with individual strands and different shades that are typical of real hair. The ear is partially visible and displays a natural contour and complexion consistent with the visible hair, suggesting that the image is of a real person. To recognize it from a bunch of deepfake tampered pictures, one would look for inconsistencies in skin texture, unnatural symmetry, or artifacts around the hair or ears—none of which are present in this image. The natural lighting and lack of any visible digital artifacts also support the conclusion that the image is authentic and has not been manipulated."

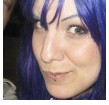

Was this image captured straight from the camera? Share your conclusion and reasons. If modifications are detected, output the tampered area's mask.

1. Whether the image has been tampered with/description of the tampered area: The image appears to be authentic and has not been tampered with. Visible parts of the image show natural features and consistent lighting.
2. Judgment basis: The visible hair shows natural color variations and texture consistent with real hair. The lighting on the hair suggests a single light source, which is coherent and creates realistic shadows and highlights. The portion of the ear visible in the image has natural contours and skin tone. The visible clothing appears to have a consistent texture and sits on the body in a way that aligns with the natural folds expected from the fabric and the pull of gravity. These details indicate that the image is likely a real photograph and would stand out as authentic among deepfake tampered pictures due to the natural inconsistencies and the coherent lighting that deepfake images often lack."

Figure 20: MMTD-Set data samples related to DeepFake tampering. The four images are all from the DFFD (Dang et al., 2020) dataset. The first two pictures have been tampered with, but the last two pictures have not been tampered with.

