# OpenReview forum: "FakeShield: Explainable Image Forgery Detection and Localization via Multi-modal Large Language Models"
_ICLR.cc/2025/Conference — ICLR 2025 Poster_

### Official Review · Reviewer_aLrU · 2024-10-30

**Soundness:** 4
**Presentation:** 4
**Contribution:** 3
**Rating:** 6
**Confidence:** 5

**Summary:**

The paper presents FakeShield, a multi-modal framework for explainable image forgery detection and localization. It uses multi-modal large language model to detect manipulations, generate tampered region masks, and provide detailed explanations based on visual artifacts and semantic inconsistencies. Utilizing carefully designed prompts, GPT-4o constructs the MMTD-Set to enhance model training. Incorporating modules like DTE-FDM and MFLM, FakeShield achieves superior performance in detection accuracy and localization across various forgery techniques compared to existing IFDL methods.

**Strengths:**

This paper possesses several notable strengths:

1.Clear motivation and remarkable innovation: This paper firstly proposes an explainable image forgery detection and localization task and innovatively designs a multi-modal large model Fakeshield, which enhances the transparency of forgery detection and effecitively pinpoints the tampered areas.

2.Cross-domain generalization and satisfactory performance: The proposed FakeShield can effectively differentiate and handle various tampering domains such as PS-tampered, DeepFake, and AIGC-based editing. The detection, explaination, and localization performance outperforms most of the SOTA methods, as reported on Tab.1, 2, 3, 4.

3.Dataset construction and potantial contribution: This paper uses GPT-4o to enrich existing datasets through carefully designed prompts, constructing the Multi-Modal Tamper Description dataSet (MMTD-Set). It has the potantial to serve for the IFDL community and future explorations.

4.Clear organization and good writing: The writing and organization of this paper is generally good, easily comprehensible, and effectively presents its methods in Figures 1-3.

**Weaknesses:**

1.Unclear statement about domain tag generator: The paper does not clearly explain how the domain-tag generator works to handle different types of image tampering. It is suggested to explain the network structure, working principle and training method of domain-tag generator more clearly.

2.Insufficient comparison with DeepFake detection methods: In Tab.2, the paper only compares FakeShield with two other DeepFake detection methods. This limited number of comparison models may not provide a comprehensive evaluation of FakeShield’s performance against a wider range of existing state-of-the-art methods. Please add more deepfake detection method for comparison.

3.Incomplete ablation studies: In Tab.6, the authors only conducted experiments on CASIA1+ dataset for PS tampering, which may have some bias. It is suggested to verify the role of domain-tag generator on more datasets.

**Questions:**

Please refer to the weakness.

---

> ### Author Response · Authors · 2024-11-21
> **Response to Reviewer aLrU**
>
> Thank you for your valuable comments! If there are any additional comments to be added, please continue the discussion with us.
>
>
>
> > **Weakness #1: Unclear Statement About Domain Tag Generator**
>
> - **Motivation and Explanations:** We observed that when training M-LLM for image tampering detection, the performance is better when the training data comes from a single data domain. However, when training data from multiple domains is directly mixed, the performance deteriorates. We attribute this to domain differences between datasets, as directly mixing training data may interfere with the model's ability to make accurate judgments. Therefore, we consider adding a textual "tag" indicating the suspected tampering type before the instructions are input into the M-LLM. For example, "This image is suspected to have been tampered with by Photoshop." Using DTG to differentiate the data domains of input images and provide distinct tags to the M-LLM reduces the impact of domain differences on the M-LLM. This approach addresses the generalization issue across different data domains and significantly enhances the practical value of FakeShield.
>
> - **Implementation Details:** We use ResNet50 to implement DTG. We randomly selected 4,000 images from each of the CASIAv2, DeepFake training set, and AIGC-Editing training set, regardless of whether they were authentic or tampered with. Each dataset was assigned a label of 0, 1, or 2, representing different data domains. We used cross-entropy loss to train the DTG, constraining it to perform a three-class classification task for the images. In Table 6, we conducted ablation experiments on the DTG. The results demonstrate that the DTG module plays a significant role in improving generalization.
>
>
>
> > **Weakness #2: Insufficient Comparison with DeepFake Detection Methods**
>
> Following your suggestion, we trained and tested two other state-of-the-art methods, Exposing [1] and RECCE [2], on our dataset. The results are shown below. It is evident that our method demonstrates superior performance compared to other competitive methods. Relevant results are presented **in Table 2 of our main paper**.
>
> |      | Exposing | RECCE | FakeShield |
> | ---- | -------- | ----- | ---------- |
> | ACC  | 0.82     | 0.92  | 0.98       |
> | F1   | 0.84     | 0.92  | 0.99       |
>
>
>
> > **Weakness #3: Incomplete Ablation Studies**
>
> Thanks for your valuable comment. Following your suggestion, we added IMD2020 to Table 6 for testing to avoid test bias. The detection accuracy results (ACC) are shown below. We also provide the complete results **in the updated Table 6 of our main paper**.
>
> |              | CASIA1+ | IMD2020 | DeepFake | AIGC-Editing |
> | ------------ | ------- | ------- | -------- | ------------ |
> | Ours w/o DTG | 0.92    | 0.71    | 0.89     | 0.72         |
> | Ours         | 0.95    | 0.83    | 0.98     | 0.93         |
>
>
>
> > **Reference**
>
> [1] Exposing the Deception: Uncovering More Forgery Clues for Deepfake Detection
>
> [2] End-to-End Reconstruction-Classification Learning for Face Forgery Detection

---

> > ### Comment · Reviewer_aLrU · 2024-11-27
> >
> > Thanks for the authors' efforts. The authors' response has addressed the concerns I raised.

---

> > > ### Author Response · Authors · 2024-11-27
> > > **Thanks Reviewer aLrU for recognizing our work**
> > >
> > > Dear Reviewer aLrU:
> > >
> > > Thank you very much for your valuable suggestions and timely responses. We truly appreciate your thoughtful feedback and the recognition of our efforts.
> > >
> > > Best Regards,
> > >
> > > Authors of #328

---

### Official Review · Reviewer_1sC4 · 2024-10-31

**Soundness:** 3
**Presentation:** 2
**Contribution:** 3
**Rating:** 6
**Confidence:** 5

**Summary:**

The paper employs LLM to enhance existing IFDL datasets from a forgery perspective. Based on these new data, a multimodal and explainable forgery detection method (FakeShield) is proposed. FakeShield first derives text description of the possible forgery type and reasons based on the finetuned LLM, and then uses the text description to guide the output of the forgery location mask. Experimental demonstrations in forgery detection, localization, and explanation aspects demonstrate the effectiveness of the proposed method.

**Strengths:**

The paper generates language descriptions for some existing IFDL datasets based on GPT-4o, so as to explain the types and causes of forgeries.

The introduction of the Domain Tag Generator offers initial predictions of the domain O_det of input images, thereby avoiding conflicts in tampered data domains and precisely guiding subsequent detection/localization models towards the forgery types of interest.

The proposal of MFLM involves the alignment of textual (O_det) and image features, facilitating the joint prediction of forgery masks from a multimodal perspective.

The experiment compares the traditional forgery detection and localization with the existing IFDL methods, and also compares with the existing LLMs from the perspective of forgery interpretability.

**Weaknesses:**

I believe there are some critical issues in the following aspects:

Dataset:
- Using GPT to generate textual descriptions for forgery regions and treating them as ground-truth (GT) lacks comparative validation. While the generation process of these textual GTs involves multiple rounds of dialogue and expert proofreading to ensure accuracy, I believe it lacks internal validation. Specifically, the mask GTs of existing datasets are (mainly) constructed based on the residual comparison between the original (authentic) images and forged ones to prevent mislabeling authentic regions as forgery. However, in creating textual GTs, this comparative process is missing, making it challenging to ensure the credibility of the textual GTs. Therefore, I suggest that GPT should simultaneously receive "original image, forged image, and/or forgery mask" for joint assessment, potentially enhancing the precision of the generated textual GTs.
- Furthermore, Figure 2 depicts the dataset construction process involving "expert proofreading", but this procedure is not mentioned in the main text.

Methodology:
- The modules (LLM, SAM, specialized token <seg>, etc.) and loss functions (ce, bce, dice, etc.) involved in the FakeShield are largely derived from existing works. Therefore, I think its original contribution is not solid enough.
- Given that the final predicted mask M_loc relies on the guidance of O_det, a natural concern arises: if O_det produces incorrect content (for example, it generates a completely authentic description for forgery data), could M_loc be significantly misled? Along this line, incorporating a corrective mechanism in TCM might better mitigate the aforementioned situation.

Experiments:
- **Lack of cross-validation**: Although FakeShield can provide detailed explanations for forgery  reasons, it appears to forcefully interpret all forgeries from preset perspectives. For example, in the last case of Figure 16, the inside of the clock is forged, while the external background/shadow, etc. are original (real). However, FakeShield still believes that its shadows do not conform to the light source logic. Therefore, it is necessary to add some forgeries that only contain some types to cross-validate whether FakeShield has truly learned the knowledge of different types of forgeries. For example, manipulating only the shadow part of an image while keeping other aspects such as lighting, edges, resolution, perspective relationships, and physical laws unchanged, to assess whether FakeShield can identify this specific manipulation involving only shadows without affecting the rest.
- **Lack of stronger generalization evaluation**: Based on the data distribution summarized in Table 7, the evaluation data is not completely cross-domain for training data. Specifically, the sources and creation processes of Deepfake and AIGC data in both training and testing are highly similar (e.g., all the AIGCs are generated by SD-inpainting). Therefore, the claim in Line 382 that FakeShield generalizes well to Deepfake and AIGC tampering is overstated. It is recommended to add completely cross-database data for evaluation, such as adding data from FFIW, CelebA-DF, DFDC, etc. for Deepfake detection.
- **Ablation experiment is not sufficient**:
  - Line 505 only evaluates the performance of FakeShield with and without DTG. However, the more critical question is why it is decided to use three forgery domain tags instead of more detailed tags? For instance, following HiFi-Net, it is possible to subdivide Photoshop-type data into splicing, pasting, and inpainting; and subdivide AIGC data into categories like “GANs” or “Diffusion” (of course I understand that all AIGC data in MMTD-Set is produced by the SD-inpainting technique). Similarly, the issue extends to the authentic domain (as per Figure 2, defining only “Scene” and “Face” as authentic tags). Could a broader range of authentic/forgery domain tags lead to performance improvements? I think it requires more in-depth discussion.
  - The ablation design in the paragraph at Line 515 is inadequate and should further evaluate the results when using only T_tag or T_img. Additionally, the experimentation in this section solely focuses on the CASIA1+ dataset, and there may be potential data bias.

Presentation:
- While the core focus of this paper is on "explainable", the main body of the experiments is still on forgery detection and localization. It may not be appropriate to put the experimental results on explanation in the appendix.

Incorrect formulas/symbols:
- Eq. (4): \hat{\mathbf{O}}_{det} and \mathbf{O}_{txt} are never defined or explained. The same problem applies to \hat{\mathbf{T}}_{tag}.
- The description of Line 318~322 (including Eq. (5)) is inconsistent with the previous text. Please confirm whether the model's predicted output is \hat{\mathbf{M}} or \mathbf{M}.
- Does the second case in Figure 14 show an incorrect response? The response is exactly the same as the first case.

Minor Issues:
- The symbol of Domain Tag Generator in Fig. 3 is a handwritten G, which should be consistent with the bold G used in the text (line 266).
- The caption of Fig. 3 describes that I_ori will be input into the Tamper Comprehension Module, but this does not match the content of the picture.
- Line 245: change "a original" to "an original"
- Line 1292: change "the first picture has glasses …" to "the second picture has glasses …"

**Questions:**

- Table 2: Why do competitors CADDM and HiFi-Net only achieve an accuracy close to random guessing? It seems unreasonable that competitors perform so poorly when both the training and testing sets are sourced from FFHQ and FaceAPP. Or, maybe it is because CADDM and HiFi-Net have not been retrained? (Lines 355-357 did not explicitly mention whether these methods were retrained).

- Figure 5: Why does the untrained FakeShield (epoch=0) already achieve an IoU of 0.40 on the CASIA1+ dataset? This result far surpasses the results of well-trained comparative algorithms, such as ManTraNet with 0.09 IoU and HiFi-Net with 0.13 IoU.

- Why is the localization of Deepfake-type data not evaluated in the experiment? From Figures 2 and 19, Deepfake data has masks that mark the forged areas, but the experimental results on Deepfake (Figures 4, 14, and Table 4) are deliberately omitted.

---

> ### Author Response · Authors · 2024-11-21
> **Response to Reviewer 1sC4 (Part 1)**
>
> Thanks for your valuable comments. We have tried our best to address the concerns you raised. If you have any further questions, please continue the discussion with us.
>
>
>
> > **Incorrect Formulas/Symbols**
>
> First, based on your suggestion, we have standardized the use of  ''\hat''  throughout the paper. In our revised version and the following rebuttal, ''\hat'' represents the ground truth, while the unmarked variable represents the model's output. For example,  $\mathbf{T}\_{tag}$, $\mathbf{O}\_{det}$, $\mathbf{y}\_{txt}$, $\mathbf{M}\_{loc}$ respectively denote the predicted value of DTG, LLM, TCM, and SAM,   $\hat{\mathbf{T}}\_{tag}, \hat{\mathbf{O}}\_{det}, \hat{\mathbf{y}}\_{txt}, \hat{\mathbf{M}}\_{loc}$ respectively denote the corresponding Ground Truth.
>
>
>
> > **Dataset Weaknesses: GPT Generated Text Description Lacks Comparative Verification**
>
> - It is challenging to obtain the corresponding original images for existing publicly available IFDL datasets. Some datasets, such as CASIAv2 and IMD2020, do not provide the original images. Our experimental setup is based on these public datasets, **making it difficult to find the original image** corresponding to each manipulated image.
>
> - Our testing results on GPT-4o indicate that when the model is provided with the following three combinations: (1) [tampered image, mask], (2) [original image, tampered image], and (3) [original image, tampered image, mask], the generated tampering explanations are **similar**, with minimal differences.
>
> - Considering the **high API cost** of GPT-4o, we chose not to input three images simultaneously. Additionally, since some datasets do not provide original images, we ultimately decided on the combination (1) [tampered image, mask].
>
>
>
> > **Dataset Weaknesses: Lack of Explanation of Expert Proofreading Process**
>
> - We employed a trained annotation team to review and refine GPT's responses based on the tampered images and corresponding masks.
>
> - The specific requirements include:  (1) Removing samples with incorrect authenticity judgments; (2) Removing samples where the answers directly mentioned the mask; (3) Removing samples where GPT refused to provide an answer or judgment; (4) Correcting statements with inaccurate descriptions of tampered region's location; (5) Correcting or removing unreasonable or factually incorrect explanations.
>
>
>
> > **Methodology Weakness:  About the Original Contribution**
>
> Our original contributions are as follows:
>
> - While LLM and SAM are existing modules, **they cannot be directly applied to the explainable IFDL task**. Our innovation lies in introducing, organically integrating, and training these existing modules into an efficient and precise model to solve the explainable IFDL task.
>
> - Specifically, we utilized carefully designed prompts with GPT-4o to construct the comprehensive and detailed MMTD-Set, introduced a domain tag generator to enhance generalization, and implemented a tamper comprehension module to improve textual understanding. **These designs and adjustments enable LLM and SAM to adapt to generalizable e-IFDL tasks with strong performance**.
>
> - Common loss functions like CE, BCE, and DICE are standard for most segmentation and detection tasks and are not presented as part of our contributions. **Using these common loss functions does not diminish our contributions.**
>
> - To our knowledge, we are the first to propose the explainable IFDL task for natural images and innovatively solve it using M-LLM and large vision models. **Our contributions have already been acknowledged by Reviewer VXsL and 1sC4**.

---

> ### Author Response · Authors · 2024-11-21
> **Response to Reviewer 1sC4 (Part 2)**
>
> > **Methodology Weaknesses: Whether the Wrong $\mathbf{O}\_{det}$ Misleads $\mathbf{M}\_{loc}$ (See Table A.3 of our main paper)**
>
> - First, our MFLM inherently possesses some error-correction capability. This is because, during training, we use potentially inaccurate $\mathbf{O}\_{det}$ as input and encourage the model to output the correct $\hat{\mathbf{M}}\_{loc}$. Thus, during testing, if $\mathbf{O}\_{det}$ contains minor inaccuracies, such as incorrect descriptions of tampered locations, MFLM can correct them.
>
> - When $\mathbf{O}\_{det}$ classifies an image as authentic, we do not input it into MFLM. This is because the detection accuracy of our DTE-FDM generally exceeds 90%, making it more reliable than MFLM.
>
> - Following your suggestion, we conducted an ablation study. During MFLM training, we used $\hat{\mathbf{O}}\_{det}$ to constrain TCM's output to correct the input $\mathbf{O}\_{det}$. The results in the table below indicate that correcting $\mathbf{O}\_{det}$ does not improve $\mathbf{M}\_{loc}$ predictions, possibly due to interference between mask and text optimization. Only the IoU results from the tests are presented here.
>
>   |                                  | CASIA1+ | IMD2020 | DeepFake | AIGC-Editing |
>   | -------------------------------- | ------- | ------- | -------- | ------------ |
>   | Using correct $\mathbf{O}_{det}$ | 0.51    | 0.49    | 0.13     | 0.12         |
>   | Ours                             | 0.54    | 0.50    | 0.14     | 0.18         |
>
>
>
> > **Experimental Weakness: Lack of Cross-validation**
>
> - In the clock example of Figure 17, FakeShield identifies inconsistencies between the shadows and lighting on the clock and those in other parts of the image. Specifically, the bright spots on the clock and the unnatural shadow along its right edge are examples of incorrect lighting and shadow conditions.
> - Moreover, FakeShield adapts its tampering explanations based on different images, selecting various angles of analysis. For instance, the first example in Figure 17, uses lighting, edges, perspective relationships, shadows, and physical laws for explanation, while the second example also includes resolution as an additional angle. Additionally, although there are eight predefined analytical angles, most responses do not provide reasons from all eight perspectives.

---

> ### Author Response · Authors · 2024-11-21
> **Response to Reviewer 1sC4 (Part 3)**
>
> > **Experimental Weakness: Lack of Stronger Generalization Evaluation**
>
> DeepFake generalization verification **(See Table 2 of our main paper)**
>
> - DeepFake manipulations include:  identity switching, facial attribute modification, and full-face generation. Among these, only facial attribute modification involves facial tampered masks, which closely align with our detection and **localization** task. To construct the MMTD-Set with a united pipeline, we only consider facial attribute modification in our work.  **In line 760 of the limitations section**, we have already stated it.
>
> - The datasets FFIW, Celeb-DF, and DFDC, primarily focus on identity switching and full-face generation, which differ from our current task setup. We will consider supporting all types of deepfake detection in future work.
>
> - To verify our DeepFake generalization ability,  we test our Fakeshield on Seq-Deepfake [1] **without additional training**, where Seq-DeepFake includes images of facial attribute manipulations generated by StyleMapGAN [2] and Talk-To-Edit [3]. The table below presents our leading performance on the unseen dataset.
>
>   |      | CADDM | HiFi-DeepFake | FakeShield |
>   | ---- | ----- | ------------- | ---------- |
>   | ACC  | 0.53  | 0.52          | 0.83       |
>   | F1   | 0.59  | 0.57          | 0.91       |
>
>
>
> AIGC-Editing generalization verification **(See A.4 of our appendix)**
>
> - Following your suggestion, we further used ControlNet Inpainting and SDXL Inpainting to generate 2,000 tampered images respectively, to verify the generalization. We selected MVSS-Net, CAT-Net, and HiFi-Net, which performed well in the AIGC-Editing data in Tables 1 and 4, as comparison methods for testing. The results are provided in the tables below.
>
> - ControlNet Inpainting data test results:
>
>   |          | ACC (Image level) | F1 (Image level) | IoU (Pixel level) | F1 (Pixel level) |
>   | -------- | ----------------- | ---------------- | ----------------- | ---------------- |
>   | MVSS-Net | 0.38              | 0.27             | 0.05              | 0.09             |
>   | CAT-Net  | 0.90              | 0.89             | 0.11              | 0.17             |
>   | HiFi-Net | 0.49              | 0.66             | 0.17              | 0.28             |
>   | Ours     | 0.99              | 0.99             | 0.18              | 0.24             |
>
> - SDXL Inpainting data test results:
>
>   |          | ACC (Image level) | F1 (Image level) | IoU (Pixel level) | F1 (Pixel level) |
>   | -------- | ----------------- | ---------------- | ----------------- | ---------------- |
>   | MVSS-Net | 0.35              | 0.20             | 0.05              | 0.08             |
>   | CAT-Net  | 0.86              | 0.83             | 0.06              | 0.09             |
>   | HiFi-Net | 0.44              | 0.61             | 0.02              | 0.03             |
>   | Ours     | 0.99              | 0.99             | 0.18              | 0.23             |
>
>
>
> > **Experimental Weaknesses: Discussion on More Detailed Domain Tags**
>
> - First, it is worth noting that the tags used by our DTG are not the five "types" labeled on the left side of Figure 2, but rather the three tags highlighted in red in Figure 3' s $\mathbf{T}_{tag}$, namely **"PhotoShop / DeepFake / AIGC."**
> - Further subdividing the "PhotoShop" tag into "splicing", "pasting", and "inpainting", or breaking down "AIGC" into finer categories, would require the DTG to directly determine whether an image has been tampered with and identify the specific tampering method. This approach essentially turns DTG into a tampering detection module, which is unreasonable and deviates from our original intention of using M-LLM to assess the authenticity of images. **Such a shift would diminish the role of the M-LLM in our method.**
> - Thus, we want DTG to **roughly classify data domains** rather than provide detailed detection answers about the specific types of manipulation. This module is intended to assist and guide the M-LLM in cross-domain generalization for tampering detection and explanations. Under this setup, the tampering detection is entirely handled by the M-LLM, which is one of the key distinctions between our method and previous works.

---

> ### Author Response · Authors · 2024-11-21
> **Response to Reviewer 1sC4 (Part 4)**
>
> > **Experimental Weaknesses: More Ablation Studies on LLM in the DTE-FDM (See Table A.3 of our main paper)**
>
> - Following your suggestion, we have added ablation experiments to investigate the roles of $\mathbf{T}\_{tag}$, $\mathbf{T}\_{img}$, and $\mathbf{T}\_{ins}$. When training MFLM, we kept other training configurations unchanged and adjusted the inputs to: {$\mathbf{T}\_{ins},\mathbf{T}\_{img}$} and {$\mathbf{T}\_{ins}, \mathbf{T}\_{tag}$}.
>
> - To avoid data bias, we conducted tests on the CASIA1+, IMD2020, DeepFake, and AIGC-Editing datasets. The IoU results are shown below. **The complete table can be found in Table 9 of the appendix.**  Using input {$\mathbf{T}\_{ins},\mathbf{T}\_{img}$} is slightly better than  {$\mathbf{T}\_{ins}, \mathbf{T}\_{tag}$}, but both are inferior to our method with the input {$\mathbf{O}\_{det}, \mathbf{T}\_{img}$}. This indicates that using an LLM to first analyze the basis for tampering and then guiding the localization achieves better results.
>
>   |                                                          | CASIA1+ | IMD2020 | DeepFake | AIGC-Editing |
>   | -------------------------------------------------------- | ------- | ------- | -------- | ------------ |
>   | {$\mathbf{T}\_{ins},\mathbf{T}\_{img}$}                    | 0.50    | 0.48    | 0.13     | 0.12         |
>   | {$\mathbf{T}\_{ins}, \mathbf{T}\_{tag}$}                   | 0.49    | 0.47    | 0.12     | 0.12         |
>   | {$\mathbf{T}\_{ins}, \mathbf{T}\_{tag}, \mathbf{T}\_{img}$} | 0.51    | 0.48    | 0.13     | 0.11         |
>   | {$\mathbf{O}\_{det}, \mathbf{T}\_{img}$} (Ours)            | 0.54    | 0.50    | 0.14     | 0.18         |
>
>
>
> > **Presentation Weakness: Incorrect formulas/symbols and Minor Issues**
>
> - Weaknesses of the Presentation:
>
>   - Due to the page limit, the explanation text is too lengthy to include in the main text.
>
>   - Thanks for your suggestion, we have added Figure 6 in the main text to present the results of the tampering explanations.
>
> - Incorrect formulas/symbols and Minor Issues:
>   - Thank you for pointing out our errors carefully. We have corrected them in the main paper and appendix.
>
>
>
> > **Question #1: Competitors of Table 2 Have Low Accuracy**
>
> - CADDM and HiFi-DeepFake are DeepFake detection methods designed for videos and cannot be trained on image datasets.
> - To ensure a fair comparison, we selected two image-based DeepFake detection sota methods, Exposing [4] and RECCE [5], and trained them with the same dataset as FakeShield. The results are shown below.  Obviously, our method far performs better than other methods in ACC and F1-score.
>
> |      | Exposing | RECCE | FakeShield |
> | ---- | -------- | ----- | ---------- |
> | ACC  | 0.82     | 0.92  | 0.98       |
> | F1   | 0.84     | 0.92  | 0.99       |
>
>
>
> > **Question #2: Figure 5 Untrained FakeShield Has Achieved High IoU**
>
> Our training epochs are numbered starting from 0, meaning the 0.40 and 0.42 values represent the IoU test results after completing the 0th epoch of training. To avoid ambiguity, we will redesign the x-axis of Figure 5 to start from 1.
>
>
>
> > **Question #3: Localization Results for DeepFake-Type Data**
>
> - Considering that most mainstream DeepFake detection methods lack tampering localization capabilities, and few studies address the "DeepFake tampering localization task," we did not initially present the localization results for DeepFake manipulations.
> - Following your suggestion, we have included the localization results for DeepFake data in Figure 4, Figure 14, and Table 4.  The localization test results for DeepFake data are shown in the table below. It can be seen that our method effectively generalizes to the DeepFake data domain.
>
> |      | SPAN | MantraNet | OSN  | HiFi-Net | PSCC-Net | CAT-Net | MVSS-Net | FakeShield |
> | ---- | ---- | --------- | ---- | -------- | -------- | ------- | -------- | ---------- |
> | IoU  | 0.04 | 0.03      | 0.11 | 0.07     | 0.12     | 0.10    | 0.10     | **0.14**   |
> | F1   | 0.06 | 0.05      | 0.13 | 0.11     | 0.18     | 0.15    | 0.09     | **0.22**   |
>
>
>
> > **Reference**
>
> [1] Detecting and Recovering Sequential DeepFake Manipulation
>
> [2] Exploiting Spatial Dimensions of Latent in GAN for Real-time Image Editing
>
> [3] Talk-to-Edit: Fine-Grained Facial Editing via Dialog
>
> [4] Exposing the Deception: Uncovering More Forgery Clues for Deepfake Detection
>
> [5] End-to-End Reconstruction-Classification Learning for Face Forgery Detection

---

> ### Comment · Reviewer_1sC4 · 2024-11-26
> **Response to Rebuttal**
>
> Thanks for the detailed responses and revisions to the issues raised. I'm generally satisfied with the current version and will reconsider the rating and discuss it with other reviewers and AC.

---

> > ### Author Response · Authors · 2024-11-26
> > **Thanks for the recognition of Reviewer 1sC4**
> >
> > Dear Reviewer 1sC4:
> >
> > Thank you for your invaluable comments, timely response, and recognition of our work. This has greatly encouraged us and made our paper more complete and coherent. If we have addressed your concerns, we earnestly look forward to you reconsidering your score. Once again, thanks for your efforts and recognition!
> >
> > Best Regards,
> >
> > Authors of #328

---

### Official Review · Reviewer_VXsL · 2024-11-04

**Soundness:** 3
**Presentation:** 3
**Contribution:** 3
**Rating:** 8
**Confidence:** 5

**Summary:**

The paper designs a multimodal framework called FakeShield, which uses the power of large language models to train the model by building a Multi-Modal Tamper Description dataSet (MMTDSet), enabling it to analyze, detect, and locate image tampering. The author uses GPT-4o to generate text descriptions and convert the existing IFDL image dataset into a dataset containing accurate text descriptions.

**Strengths:**

1. FakeShield is the first proposed multimodal large-scale image forgery detection and localization model, which can provide a more reasonable basis for judgment.
2. By combining visual and language models, the paper's method can provide more comprehensive image analysis. The paper enhances the existing IFDL dataset through GPT-4o and constructs MMTDSet, which also makes a certain contribution to the field of forgery detection.
3. The DTEFDM and MFLM modules proposed in the paper enable the model to flexibly handle different types of image tampering. This paper has potential value in practical applications.

**Weaknesses:**

1. The model proposed in the paper may require high computing resources and training time.
2. The generalization ability of the model on new types of tampering that have not been seen yet needs further verification.  Also further work may be needed to quantify and verify the quality of the model's explanation.

**Questions:**

It is recommended that the authors further validate model generalizability as well as quantify and validate the quality of the model explanations.

---

> ### Author Response · Authors · 2024-11-21
> **Response to Reviewer VXsL (Part 1)**
>
> Thank you for your constructive comments! We hope that our response will address all of your concerns. All discussions and supplementary analyses will be included in our revised version. If there are any additional comments to be added, please continue the discussion with us.
>
>
>
> > **Weakness #1: High Computing Resources and Training Time**
>
> - With some quantitative fine-tuning technologies (e.g. QLoRA, etc.) and a small batch size, our model can even be trained on a single NVIDIA GTX 3090Ti GPU (24GB), which requires relatively low computational resources.
>
> - On four NVIDIA A100 40GB GPUs, the training time for DTE-FDM is only **5 hours**, and for MFLM, it is just **10 hours**. Furthermore, since the model only needs to be trained once for continuous use, this duration and resource consumption are acceptable.
>
>
>
> > **Weakness #2: Verifying Generalization to New Types of Tampering**
>
> Our training set includes tampered data constructed using stable diffusion inpainting and FaceApp. We consider expanding the testing set with more AIGC-Editing data generated via ControlNet Inpainting and SDXL Inpainting, as well as DeepFake data manipulated with StyleMapGAN [1] and Talk-To-Edit [2]. **New results are included in Table 2 and Table 10 of our main paper.**
>
>
>
> More AIGC-Editing test data:
>
> - We use new tampering methods that have not been seen in models (ControlNet Inpainting and SDXL Inpainting) to construct 2000 tampered images each to expand our test set.
>
> - We selected MVSS-Net, CAT-Net, and HiFi-Net, which performed well in the AIGC-Editing data domain in Tables 1 and 4, as comparison methods for testing. The evaluation metrics include image-level ACC and F1-score (to validate detection results) and pixel-level IoU and F1-score (to validate localization results). The test results are shown in the table below. Our method achieves leading performance across unseen datasets and exhibits good generalization.
>
> - ControlNet Inpainting data test results:
>
>   |          | ACC (Image level) | F1 (Image level) | IoU (Pixel level) | F1 (Pixel level) |
>   | -------- | ----------------- | ---------------- | ----------------- | ---------------- |
>   | MVSS-Net | 0.38              | 0.27             | 0.05              | 0.09             |
>   | CAT-Net  | 0.90              | 0.89             | 0.11              | 0.17             |
>   | HiFi-Net | 0.49              | 0.66             | 0.17              | 0.28             |
>   | Ours     | 0.99              | 0.99             | 0.18              | 0.24             |
>
> - SDXL Inpainting data test results:
>
>   |          | ACC (Image level) | F1 (Image level) | IoU (Pixel level) | F1 (Pixel level) |
>   | -------- | ----------------- | ---------------- | ----------------- | ---------------- |
>   | MVSS-Net | 0.35              | 0.20             | 0.05              | 0.08             |
>   | CAT-Net  | 0.86              | 0.83             | 0.06              | 0.09             |
>   | HiFi-Net | 0.44              | 0.61             | 0.02              | 0.03             |
>   | Ours     | 0.99              | 0.99             | 0.18              | 0.23             |
>
>
>
> More DeepFake test data:
>
> - We conducted more tests on the Seq-Deepfake [3] dataset, which includes images of facial attribute manipulations generated by StyleMapGAN and Talk-To-Edit.
>
> - The table below presents the test results on the Seq-Deepfake dataset. Our method achieved leading performance even on unseen datasets.
>
>   |      | CADDM | HiFi-DeepFake | FakeShield |
>   | ---- | ----- | ------------- | ---------- |
>   | ACC  | 0.53  | 0.52          | 0.84       |
>   | F1   | 0.59  | 0.57          | 0.91       |

---

> ### Author Response · Authors · 2024-11-21
> **Response to Reviewer VXsL (Part 2)**
>
> > **Weakness #2: Quantifying the Quality of Model's Explanation**
>
> We considered using GPT-assisted Evaluation and User Study to quantify the quality of model's explanations:
>
>
>
> GPT-assisted Evaluation
>
> - We use GPT-4o as a judge to evaluate the outputs of the five models listed in Table 3.  We combined all test results from each model and randomly selected 1,000 output samples for quantitative evaluation.
>
> - The images, GT masks (if available), and model outputs will be provided to GPT-4o. It will assess the outputs based on four dimensions: (1) detection accuracy, (2) accuracy of tampered region descriptions, (3) reasonableness of explanations, and (4) level of detail. A total score from 0-5 will be given, where higher scores indicate better performance. Samples with incorrect detection results or model refuse to answer will be directly scored as 0.
>
> - We calculated the average score for each model, and the results are shown in the table below. Our method outperforms all comparison methods.
>
>   | Method        | GPT-4o | LLaVA-v1.6-34B | InternVL2-26B | Qwen2-VL-7B | FakeShield |
>   | ------------- | ------ | -------------- | ------------- | ----------- | ---------- |
>   | Average Score | 3.10   | 2.63           | 2.76          | 2.41        | **4.43**   |
>
>
>
> User Study
>
> - We invited 40+ volunteers to provide user reviews for the outputs of the five models listed in Table 3. We combined samples from all datasets and randomly selected 15 images.
>
> - These 15 images, along with their corresponding masks (if available) and the outputs of the five models, were included in the questionnaire. Users were asked to rate the outputs with an overall score from 0 to 5 based on four criteria: (1) detection accuracy, (2) accuracy of tampered region descriptions, (3) reasonableness of explanations, and (4) level of detail.
>
> - The table below summarizes the average score for each model. Our method achieves the best performance among all compared methods.
>
>   | Method        | GPT-4o | LLaVA-v1.6-34B | InternVL2-26B | Qwen2-VL-7B | FakeShield |
>   | ------------- | ------ | -------------- | ------------- | ----------- | ---------- |
>   | Average Score | 3.67   | 3.02           | 3.26          | 3.17        | **4.80**   |
>
>
>
> > **Reference**
>
> [1] Exploiting Spatial Dimensions of Latent in GAN for Real-time Image Editing
>
> [2] Talk-to-Edit: Fine-Grained Facial Editing via Dialog
>
> [3] Detecting and Recovering Sequential DeepFake Manipulation

---

### Public Comment · ~Tyler_Lucas_Nuremberg1 · 2024-11-14
**Question about the Performance of Deepfake Detection Baselines with FaceApp-Generated Images**

The deepfakes in the proposed dataset, which are derived from the DFFD, are generated by FaceApp (2017).
Given that this face-swapping method is no longer particularly cutting-edge, it is surprising to see state-of-the-art detectors like CADDM struggling with binary classification between real images and those altered by FaceApp, as indicated in Table 2.

This might suggest issues with the experimental setup or perhaps a misunderstanding on my part. A more detailed explanation from the authors would be helpful.

---

> ### Author Response · Authors · 2024-11-15
>
> Thank you for your attention. We will provide a detailed explanation in our subsequent rebuttal.

---

### Public Comment · ~Zhengchao_Huang1 · 2024-11-15
**Some suspected plagiarism traces**

Upon reviewing this paper, we have identified significant overlaps with prior work that do not appear to be properly cited or acknowledged. Specifically, it appears that content from our prior work has been substantially copied into this paper.

**Source Paper**
* Title: **FFAA: Multimodal Large Language Model based Explainable Open-World Face Forgery Analysis Assistant**
* Link: https://arxiv.org/abs/2408.10072

**Overlaps**
1. **Similar task scenario and methodology:** Both papers operate in related fields. The methodologies in both articles involve augmenting existing datasets using GPT-4o, then distilling the knowledge to MLLM, and completing the assigned tasks through semantic guidance.
2. **Serious plagiarism in the dataset construction pipeline:** The diagrams (Fig.2) are strikingly similar, with nearly identical processes and similar prompt designs. The Fakeshield article should have mentioned how FFAA’s method constructs the dataset and what improvements Fakeshield made, instead of claiming it as their own contribution.
3. **Similar writing structure:** Both papers start from similar motivations, apply nearly identical methodologies, and follow essentially the same dataset construction process. The appendix also shows significant overlap.
4. **Diagrams:** The color schemes, pipeline designs, and other elements are highly similar, especially Fig 2.

These issues indicate clear violations of the ethical standards and submission guidelines set forth by ICLR and other scientific publication bodies.

---

> ### Author Response · Authors · 2024-11-15
> **Firmly Deny Plagiarism of the FFAA Paper**
>
> We completely $\textcolor{red}{\textbf{reject}}$ the accusation that we plagiarized the FFAA paper. FakeShield and FFAA have entirely different insights, methods, tasks, and experimental results. Below are our responses to your concerns, point by point.
>
>
>
> > **Writing Structure and Composition**
>
> - Since LLaVA [1], the paradigm of using GPT-4/GPT-4o to label data has been widely adopted and well-known [2, 3]. We just adopted this paradigm, and our Figure 2 simply referenced and integrated some elements and partial layouts from FFAA, ShareGPT4v [2] and I2EBench [3].  In fact, Figure 2 of FFAA and Figure 3 of ShareGPT4v [2] are highly consistent in terms of color schemes and layout, as well as FFAA's Figure 13 and ShareGPT4v's Figure 11. **Thus, the layout of Figure 2 is not original to FFAA.**  **Anyway, we will cite FFAA paper in the revised version and introduce FFAA approach.**
>
>
>
> > **Different Task Settings**
>
> - FFAA focuses solely on deepfake detection for faces, while our method primarily focuses on detection and **localization** tasks for natural images, and can be generalized to deepfake and AIGC detection. Therefore, the types of manipulations addressed in our paper are more diverse, and the tasks we aim to accomplish are more complex.
>
>
>
> > **Significant Methodological Differences**
>
> - In dataset construction, although both FFAA and our work use GPT-4o, we have provided more detailed and varied descriptions for different types of manipulations, including edge artifacts, lighting changes, perspective relationships, etc.  Using GPT-4o is not our contribution. **The core of our dataset construction** lies in the prompt design, as well as the significant human effort and GPT-4o token costs involved in data collection and organization.
>
>
>
> - In methodology, we have innovatively proposed a domain-tag guided detection strategy, which enhances the generalization ability of M-LLM. Additionally, for localization, we introduced a tamper comprehensive module to guide SAM to correctly segment the mask based on textual descriptions. **These methods are completely different from those used in FFAA.**
>
>
>
> > **Completion Timeline and the Proof of Our Originality**
>
> - Our work began around April of this year and our dataset construction was completed in June, while FFAA was published on the arXiv platform on August 19th. Therefore, there was no influence from FFAA on the construction of our ideas, the determination of methods, or the experimental setup.
> - **We will provide related proof materials, including screenshots of group meeting discussion videos, to the PC, and will make these publicly available after the review process concludes.**
>
>
>
> In conclusion, $\textcolor{red}{\textbf{we firmly deny plagiarism of the FFAA paper}}$. We believe our method demonstrates significant innovation, substantial effort, and potential contributions to the community. We have faithfully adhered to the submission principles of the ICLR conference and respect the research achievements of all authors. We trust that the Program Committee and reviewers will provide a fair evaluation of our work.
>
>
> > **Reference**
>
> [1] Visual Instruction Tuning, https://arxiv.org/abs/2304.08485.
>
> [2] ShareGPT4V: Improving Large Multi-modal Models with Better Captions, https://arxiv.org/pdf/2311.12793.
>
> [3] I2EBench: A Comprehensive Benchmark for Instruction-based Image Editing, https://arxiv.org/abs/2408.14180.

---

### Author Response · Authors · 2024-11-21
**Global Response**

Thank you to all the reviewers for their constructive feedback. All reviewers have acknowledged and praised our innovation, methodology, and experimental results. We have addressed all the concerns, responded to each point, and made revisions to the paper. The modified sections are marked in $\textcolor{red}{\text{red}}$. **We sincerely hope you recheck and reconsider your decision.**

---

### Author Response · Authors · 2024-11-25
**Looking Forward to Further Discussions**

Dear reviewers:

Thanks a lot for your previous constructive comments. We would like to know if our revisions have addressed your concerns? We welcome any discussions and suggestions that will help us further improve this paper.

Best Regards,

Authors of #328

---

### Meta-Review · Area_Chair_uhSv · 2024-12-21

**Metareview:**

1x accept, 2x borderline accept. This paper introduces a multimodal large-language-model-based approach to detect, localize, and explain image forgeries by constructing and leveraging a new multimodal dataset enriched with GPT-generated textual descriptions. The reviewers agree on the (1) effective combination of visual and textual cues for detection and explanation, (2) strong cross-domain performance against various tampering types, and (3) clear potential for practical applications in forgery analysis. However, they note (1) potentially high computational overhead, (2) the need for stronger evidence of generalization to unseen tampering domains, and (3) the importance of thoroughly quantifying the explanation quality. The authors’ follow-up responses address these concerns through detailed clarifications on resource requirements, extensive additional experiments on new data, and user/GPT-based evaluations of the explanations, so the AC leans to accept this submission.

**Additional Comments On Reviewer Discussion:**

N/A

---

### Decision · Program_Chairs · 2025-01-22

Accept (Poster)